# $\alpha$-GFN: GENERALIZED MIXING IN GFLOWNETS FOR BETTER EXPLORATION-EXPLOITATION TRADE-OFF

## ABSTRACT

Standard Generative Flow Network (GFlowNet) training implicitly assigns equal weights to the forward and backward policies, a consequence of the flow-matching view that constrains the exploration–exploitation dynamics. Extending the connection between GFlowNets and Markov chains, we show that this equal weighting arises from a theoretical equivalence between GFlowNet objectives and Markov chain reversibility. Building on this, we introduce **$\alpha$-GFNs**, which generalize standard GFlowNet training from strictly balanced flows to imbalanced flows by mixing the forward and backward policies with a hyperparameter $\alpha$ in the training objectives. Through the link to reversibility, we further establish that such objectives converge to unique flows. This generalization provides a richer exploration–exploitation trade-off and, in some settings, coarse control over trajectory lengths. We also propose a simple scheduling algorithm to combine the strengths of different $\alpha$ values. Experiments on Set Generation, Bit Sequence Generation, and Molecule Generation demonstrate consistent performance gains and highlight the benefits of $\alpha$ tuning.

## 1 INTRODUCTION

Generative Flow Networks (GFlowNets) (Bengio et al., 2021) are generative models that sample compositional objects from high-dimensional distributions with probabilities proportional to a reward function (Bengio et al., 2021). They are sampling methods that originate from the intersection of reinforcement learning frameworks (Tiapkin et al., 2024; Mohammadpour et al., 2024; Deleu et al., 2024) and flow networks (Bengio et al., 2023), offering an alternative to traditional approaches such as Markov Chain Monte Carlo (MCMC) (Brooks,

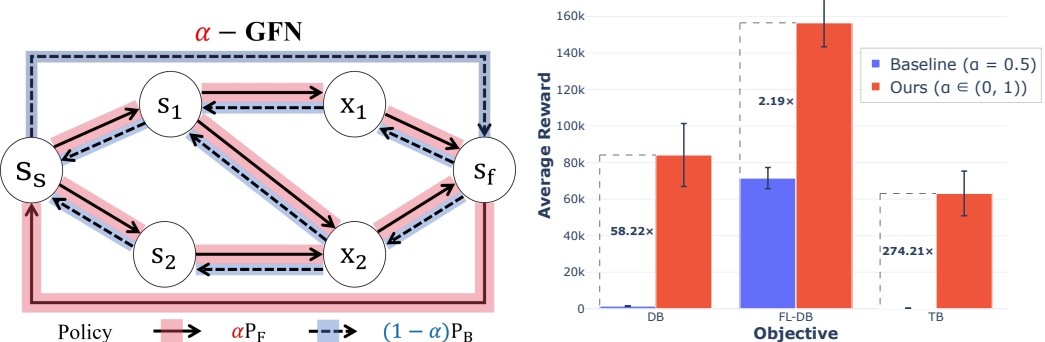

Figure 1: **(Left)** Illustration of $\alpha$-GFNs, where weights $\alpha$ and $1 - \alpha$ are assigned to forward policy $P_F$ and backward policy $P_B$, respectively. **(Right)** Enhanced performance of $\alpha$-GFN objectives in large sets (Pan et al., 2023). Objectives with unequal weights surpasses vanilla ones.

1998). Since their introduction, GFlowNets have been applied in various domains including molecular discovery (Bengio et al., 2021; Jain et al., 2023; Zhu et al., 2023), diffusion models (Zhang et al., 2024; Venkatraman et al., 2024; Liu et al., 2024) and large language models (Hu et al., 2024; Song et al., 2024; Yun et al., 2025; Zhu et al., 2025), demonstrating both mode-discovering and diversity-preserving abilities.

Despite their empirical success, most previous applications of GFlowNets implicitly assume fixed, equally weighted forward and backward transitions. This stems from the flow network perspective in GFlowNet theories (Bengio et al., 2021; 2023), despite their original formulation as Markov Decision Processes (MDPs) over pointed directed acyclic graph (DAG) state spaces. As a consequence, GFlowNet training objectives largely employ flow-matching (Bengio et al., 2021; 2023; Malkin et al., 2022; Madan et al., 2023; Pan et al., 2023), which inherently enforces equal transition weighting due to flow properties. Nevertheless, the equal weighting may not be ideal. Corresponding to imbalanced flows, unequal weights can enhance to the performance of GFlowNets, as illustrated in Fig. 1.

On the theory side, there are limited attempts to extend the theoretical framework of GFlowNets. Despite the link to reinforcement learning (Tiapkin et al., 2024; Deleu et al., 2024), GFlowNets' connections to Markov chains (MCs) remain relatively less explored. Previous work (Deleu & Bengio, 2023) has proposed linking Flow Matching (FM, Bengio et al. (2021)) objectives in GFlowNets to invariant measures in MCs. While this approach reflects certain properties of FM, the related conclusions do not generalize well to other objectives (Bengio et al., 2023; Malkin et al., 2022; Madan et al., 2023) where two policies are potentially mixed with equal weights.

In this work, we extend the theoretical link between GFlowNets and Markov chains to multiple training objectives. From this perspective, we propose a generalization of GFlowNets, termed $\alpha$-GFNs. Compared to standard GFlowNet training, our formulation introduces a hyperparameter $\alpha \in (0, 1)$ that enables flexible mixing of forward and backward policies, leading to imbalanced flows. Our theoretical analysis shows that $\alpha$-GFN objectives converge to unique flow functions. Moreover, we establish that $\alpha$ provides GFlowNets with a flexible exploration–exploitation trade-off and, in some tasks, a mechanism to control trajectory lengths. We also propose a scheduling algorithm that leverages different $\alpha$ values during training to combine their strengths. Extensive experiments on three benchmarks (Set Generation, Bit Sequence Generation, and Molecule Generation) demonstrate consistent improvements of $\alpha$-GFNs over baseline methods.

**Contributions.** We summarize the main contributions of this paper as follows:

- We introduce **$\alpha$-GFNs**, augmenting GFlowNet objectives with a hyperparameter $\alpha$ that enables flexible mixing of forward and backward policies (Def. 4.3). We further propose a tentative **scheduling algorithm** to combine the benefits of different $\alpha$ during training (Sec. 4.3).
- We extend the theoretical connections between GFlowNets and **Markov chains**, revealing the implicit equal weighting of forward and backward policies in vanilla GFlowNet objectives (Prop. 4.1, Thm. 4.2). Building on this, we show the **theoretical convergence** of $\alpha$-GFN losses to unique flow functions (Prop. 4.4), and we explain the **effectiveness** of $\alpha$-GFN losses by connecting their gradients with the gradients of vanilla GFlowNet losses (Prop. 4.5).
- We provide extensive experiments showing how $\alpha$ enables a flexible **exploration-exploitation trade-off** (Fig. 2), leading to the performance gains of our approach across multiple tasks (Sec. 5). We also show the trajectory length-controlling effect of our method in certain scenarios (Fig. 5).

## 2 PRELIMINARIES

**Generative Flow Network (GFlowNet, GFN) preliminaries.** A GFlowNet is specified by a tuple $(G, R, F, P_F, P_B)$. Here $G = (S, \mathbb{A})$ is a pointed directed acyclic graph with source $s_s$ and sink $s_f$, $R : \mathcal{X} \subseteq S \to \mathbb{R}_+$ is a reward, $F : S \to \mathbb{R}_+$ is a state flow; and $P_F, P_B : \mathbb{A} \to [0, 1]$ are forward and backward policies. Let $\mathfrak{T}^{\text{flow}}$ be the set of complete trajectories $\mathsf{t}^f = (s_0, \ldots, s_N)$ with

$s_0 = s_s$ and $s_N = s_f$. Samples are states $x \in \mathcal{X}$ sequentially constructed by $P_F$ and deconstructed by $P_B$. The generation of a sample terminates when $s_f$ is reached, yielding a complete trajectory $(s_0 = s_s, \ldots, s_{N-1} = x, s_N = s_f)$. Ideally, the probability of generating $x$ is proportional to its reward, $P_F^\top(x) \triangleq \sum_{\mathfrak{t} \in \mathfrak{T}^{\text{flow}} : x \in \mathfrak{t}} \prod_{i=1}^N P_F(s_i \mid s_{i-1}) \propto R(x)$. Under the convergence of its training objectives, this proportionality is satisfied and the uniqueness of flows is achieved.

**GFlowNet training objectives and variants.** Several losses have been introduced to train GFlowNets by enforcing flow-balance conditions, such as Flow Matching (FM, Bengio et al. (2021)), Detailed Balance (DB, Bengio et al. (2023)), Subtrajectory Balance (SubTB, Madan et al. (2023)), or Trajectory Balance (TB, Malkin et al. (2022)). In this paper, we consider mainly SubTB, which unifies DB and TB, and its variants. Given any partial trajectory $\mathfrak{t}' = (s_k, s_{k+1}, \ldots, s_{k+m}) \subset \mathfrak{t}^f \in \mathfrak{T}^{\text{flow}}$, SubTB aims at ensuring that

$$F(s_k) \prod_{i=1}^m P_F(s_{k+i} \mid s_{k+i-1}) = F(s_{k+m}) \prod_{i=1}^m P_B(s_{k+i-1} \mid s_{k+i}), \tag{1}$$

with $P_B(x \mid s_f) = \frac{R(x)}{F(s_f)}$ for all $x \in \mathcal{X}$ and $F(s_f) = \sum_{x' \in \mathcal{X}} R(x') = F(s_0)$. The loss is the log-square of Eq. 1, and a convex combination of it, termed SubTB($\lambda$), is used for training. For more details and formal definitions, see App. A.1. Although these objectives balance flows, credit assignment can be inefficient. Forward-looking (FL) variants with intermediate rewards (Pan et al., 2023; Jang et al., 2024) address this by modifying the balance with reparameterization of flows. With an intermediate energy $\mathcal{E} : \mathbb{A} \to \mathbb{R}$, SubTB becomes FL-SubTB by multiplying the right hand side of Eq. 1 with $\prod_{i=1}^m e^{-\mathcal{E}(s_{k+i-1}, s_{k+i})}$.

**Reversibility.** Reversibility is a classical concept in Markov chain theory, see e.g. (Douc et al., 2018). Given a probability measure over the state space $\pi : S \to [0, 1]$ and the transition kernel of the chain $P : S \times S \to [0, 1]$, the reversibility of $P$ implies that, at any sequence of states $(s_k, s_{k+1}, \ldots, s_{k+m})$

$$\pi(s_k) \prod_{i=1}^m P(s_{k+i} | s_{k+i-1}) = \pi(s_{k+m}) \prod_{i=1}^m P(s_{k+i-1} | s_{k+i}). \tag{2}$$

## 3 RELATED WORKS

**GFlowNet theories and connections with Markov chains.** Initially formalized within the flow network framework (Bengio et al., 2021), subsequent developments (e.g., the detailed balance conditions (Bengio et al., 2023)) have drawn heavily on Markov chain (MC) theory, underscoring the need for a rigorous link to classical MC theory. Deleu & Bengio (2023) proposed a GFlowNet-induced MC under the FM objective using only the forward policy; yet the connection with the backward policy is not mentioned whereas it is central to the connection we establish. Furthermore, Deleu & Bengio (2023) do not discuss the connection with other loss functions, while we consider several losses, including DB, SubTB, TB and their FL variants.

**GFlowNet objective design.** Standard GFlowNet losses, such as DB (Bengio et al., 2023), SubTB (Madan et al., 2023) and TB (Malkin et al., 2022), optimize with both the forward and backward policies to ensure balanced flows, which equally weights the two policies, whereas we consider more general weights in this work. Their variants include temperature-conditional GFlowNets (Zhang et al., 2023; Zhou et al., 2024; Kim et al., 2024) where a parameter is applied to scale the reward, and forward-looking GFlowNets (Pan et al., 2023; Jang et al., 2024) where intermediate energies are introduced at the transitions, and Hu et al. (2025) alters the log-square form of the losses. All of these contributions remain within the flow-matching paradigm, but our approach departs from the strict balance condition by mixing forward and backward policies via a tunable parameter, thereby allowing controlled flow imbalance.

## 4 GENERALIZING GFLOWNET TRAINING WITH $\alpha$-GFN OBJECTIVES

This section is organized into four parts. First, we show that the target of the vanilla GFlowNet objectives corresponds to the reversibility condition of a Markov chain with the **equally mixed policy** $P_{0.5} = \frac{1}{2}P_F + \frac{1}{2}P_B$ as its transition kernel. Next, we extend the mixing to unequal weights, leading to the $\alpha$-GFN objectives, whose convergence is further clarified through the underlying MC formulation. We then introduce a scheduling algorithm that combines the advantages of different $\alpha$ values. Finally, we illustrate the flexible exploration–exploitation trade-off enabled by $\alpha$.

### 4.1 THE EQUALLY MIXED MARKOV CHAIN BEHIND VANILLA GFLOWNET OBJECTIVES

We begin with an intuitive comparison of Eq. 1 and Eq. 2, which reveals a structural similarity between GFlowNets objectives and the reversibility of Markov chains. Specifically, both equations share the following structure: a sequence of transitions is coupled with a probability measure on each side. Building on the equivalence between flows and probability measures (see Deleu & Bengio (2023) and Eq. 18), this resemblance actually suggests a close connection between GFN objectives and MC reversibility even though GFNs only use the forward policy $P_F$ to generate samples. However, the policies on the two sides of Eq. 1 are not identical, which prevents a direct correspondence with Eq. 2. In reality, this obstacle can be overcome by introducing the equally mixed policy $P_{0.5}$. When applied to Eq. 2, $P_{0.5}$ naturally separates $P_F$ and $P_B$ onto different sides, eliminating the weights and recovering Eq. 1:

**Proposition 4.1.** *The reversibility of $P_{0.5}$ means for any partial trajectory $\mathfrak{t}' = (s_k, \ldots, s_{k+m})$, $\pi(s_k) \prod_{i=1}^{m} P_F(s_{k+i}|s_{k+i-1}) = \pi(s_{k+m}) \prod_{i=1}^{m} P_B(s_{k+i-1}|s_{k+i})$.*

Applying Prop. 4.1 to DB, SubTB and TB, we extend the GFN-MC link in Deleu & Bengio (2023) from FM to these objectives, and turn the similarity into a theoretical equivalence between GFNs and MCs:

**Theorem 4.2.** *The target of SubTB is equivalent to the partial-trajectory-level reversibility of a MC with the transition kernel $P_{0.5}$, and that of DB or TB is similar. Moreover, their convergence to unique flows is related to corresponding properties in MC theory.*

The proofs for Prop. 4.1 and Thm. 4.2 appear in App. C.1 and App. C.2, respectively. App. B further clarifies link from Markov chains to GFlowNets (MC→GFN), which is not elucidated in (Deleu & Bengio, 2023).

### 4.2 $\alpha$-GFN OBJECTIVES: FLEXIBLE MIXING OF POLICIES WITH A HYPERPARAMETER $\alpha$

Now, it is clear that the objectives of GFlowNets adopt an equal weight mixing of $P_F$ and $P_B$. However, the equal weights may not be ideal for all the settings. By analogy with the way two policies can be mixed by taking a convex combination of the probability distributions, we propose a flexible mixing regime with a hyperparameter $\alpha \in (0, 1)$ as the mixing ratio. To be specific, by plugging an **arbitrarily mixed policy** $P_\alpha = \alpha P_F + (1 - \alpha)P_B$ into Eq. 1, Prop. 4.1, and Thm. 4.2, we obtain objectives corresponding to reversibility of $P_\alpha$, termed $\alpha$-GFN objectives. The following definition provides the intuition; see App. A.1 for the rigorous $\alpha$-GFN loss functions.

**Definition 4.3** ($\alpha$-GFN objectives). Given $\alpha \in (0, 1)$, the $\alpha$-SubTB loss aims at ensuring that

$$\alpha^m F(s_k) \prod_{i=1}^{m} P_F(s_{k+i}|s_{k+i-1}) = (1 - \alpha)^m F(s_{k+m}) \prod_{i=1}^{m} P_B(s_{k+i-1}|s_{k+i}) \quad (3)$$

for any partial trajectory $\mathfrak{t}' = (s_k, \ldots, s_{k+m})$. Applying this hyperparameter to other variants, such as forward-looking ones, yields similar objectives, specifically, $\alpha$-FL-SubTB aims at ensuring that

$$\alpha^m F(s_k) \prod_{i=1}^{m} P_F(s_{k+i}|s_{k+i-1}) = (1 - \alpha)^m F(s_{k+m}) \prod_{i=1}^{m} P_B(s_{k+i-1}|s_{k+i})e^{-\mathcal{E}(s_{k+i-1}, s_{k+i})}. \quad (4)$$

Other losses are defined similarly. Although the targets in Def. 4.3 violate the balance condition of vanilla GFlowNets, their **convergence** to unique flows is described by the link to MC reversibility (proof in App. C.3):

**Proposition 4.4.** *The targets of $\alpha$-SubTB is equivalent to the partial-trajectory-level reversibility of a MC with the transition kernel $P_\alpha$, and those of $\alpha$-DB, $\alpha$-TB and the variants are similar. Moreover, their convergence to unique flows is similar to vanilla GFN objectives for all $\alpha \in (0, 1)$.*

### 4.3 SCHEDULING OF $\alpha$

Fixing the value of $\alpha$ may lead to undesirable effects. As shown in Fig. 3, certain choices of $\alpha$ negatively affect the reward-fitting ability of the forward policy. To address this, we propose a two-stage scheduling algorithm that retains the benefits of $\alpha$-GFNs while preserving the fitting behavior of vanilla GFNs: (i) start with $\alpha$ far from 0.5 (e.g., 0.1–0.4 or 0.6–0.9) and keep it fixed for a set number of steps; (ii) gradually anneal $\alpha$ to 0.5 over the remaining steps. See Alg. 1 for details.

---

**Algorithm 1** Scheduled Training of $\alpha$-GFNs

1: Initialize $\alpha = \alpha_0$, select the vanilla objective $\mathcal{L}$ and augment it with $\alpha$ to be an $\alpha$-GFN objective $\mathcal{L}_\alpha$.
2: Select total steps $N$, stage 1 steps $N_1$ and the annealing function $f$ at stage 2.[1]
3: Initialize model $\theta$, forward and backward policy $P_F^\theta, P_B^\theta$, flow function $F^\theta$.
4: **for** $n = 1, \ldots, N$ **do**
5:   **if** $n > N_1$ **then**
6:     $\alpha \leftarrow f(\alpha_0, n, N_1, N)$
7:   **end if**
8:   Update $\theta$ by minimizing $\mathcal{L}_\alpha(P_F^\theta, P_B^\theta, F^\theta)$.
9: **end for**

---

### 4.4 FLEXIBLE EXPLORATION-EXPLOITATION TRADE-OFF WITH $\alpha$

How does $\alpha$ contribute to the training of GFlowNets? MC theory suggests that convergence rates of $\alpha$-GFN objectives to unique flows may vary exponentially for different $\alpha$ values (see App. B.3). Although GFNs are not optimized via MCMC, this property still has significant impact on the behavior of the forward policy.

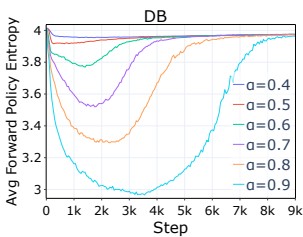 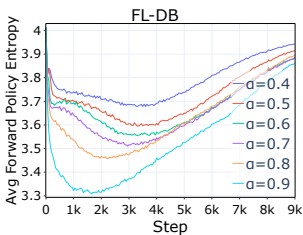 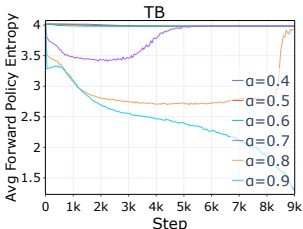

Figure 2: $\alpha$-GFN entropy mechanisms in large sets with **(Left)** DB, **(Center)** FL-DB, **(Right)** TB objectives.

The hyperparameter $\alpha \in (0, 1)$ is the mixing ratio in $P_\alpha = \alpha P_F + (1 - \alpha)P_B$, which controls the contribution of the forward policy $P_F$. In practice, $\alpha$ scales the training pressure on $P_F$: larger values of $\alpha$ accelerate exploitation of current estimates, while smaller values temper it. Combined with the GFlowNet target $P_F^\top(x) \propto R(x)$ and the convergence rates of different MCs, this leads to the following heuristic. When $\alpha > 0.5$, exploitation dominates, quickly suppressing low-reward or unseen actions and concentrating mass on high-reward ones. When $\alpha < 0.5$, exploitation slows down, which sustains broader exploration and produces a flatter action distribution. Empirically, the entropy dynamics in Fig. 2 follow this pattern: larger $\alpha$ induces an early drop in per-action entropy, while smaller $\alpha$ maintains higher entropy. This behavior can be explained by examining the gradient more closely:

**Proposition 4.5** (Gradient of $\alpha$-GFN objectives, proof in App. C.4)**.** *We take SubTB as an example and denote $P_F(\mathfrak{t}') = \prod_{i=1}^m P_F(s_{k+i} \mid s_{k+i-1})$ for simplicity. Given the fact that GFN losses are log-square*

---

[1]$f$ is a scheduling function, e.g., $f(\alpha_0, n, N_1, N) = 0.5 + (\alpha_0 - 0.5)\exp\left(-4 \cdot \frac{n - N_1}{N - N_1}\right)$.

*functions of their targets, the gradient of $\alpha$-SubTB loss can be expressed as a modification of the SubTB loss gradient for $P_F$ as:* $\frac{\partial L_{\alpha-SubTB}(\mathfrak{t}')}{\partial P_F(\mathfrak{t}')} = \frac{\partial L_{SubTB}(\mathfrak{t}')}{\partial P_F(\mathfrak{t}')} + \frac{2m}{P_F(\mathfrak{t}')} \log \frac{\alpha}{1-\alpha}.$

For $\alpha > 0.5$, the term $\frac{2m}{P_F(\mathfrak{t}')} \log \frac{\alpha}{1-\alpha}$ is larger when $P_F(\mathfrak{t}')$ is small (low reward) and smaller when $P_F(\mathfrak{t}')$ is large (high reward). Consequently, low-reward probabilities decay faster while high-reward ones decay more slowly, sharpening the distribution and strengthening exploitation. For $\alpha < 0.5$, $\log \frac{\alpha}{1-\alpha} < 0$ reverses the effect, which yields a flatter distribution and contributes to exploration. In addition, Proposition 4.5 shows that the exploration-exploitation effects scales with $\left| m \log \frac{\alpha}{1-\alpha} \right|$.

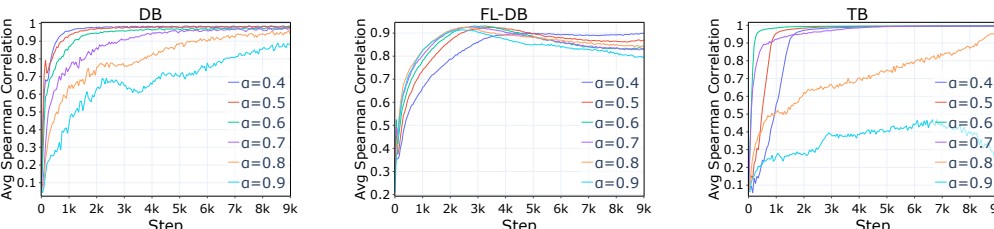

Figure 3: $\alpha$-GFN Spearman correlations in large sets with **(Left)** DB **(Center)** FL-DB **(Right)** TB objectives.

Nevertheless, training $P_F$ under the $\alpha$-GFN objectives may suffer from over-exploitation when $\alpha$ is too large or inefficient credit assignment when $\alpha$ is too small. The former is reflected by a drop in the correlation between $P_F$ and the comparison of reward in Fig. 3, and the latter is reflected by low reward of vanilla GFN objectives in Fig. 1. Thus, we schedule $\alpha$ with Alg. 1 to combine the advantages of different $\alpha$ values and avoid the undesirable effects of a fixed $\alpha$.

## 5 EXPERIMENTS

Previous sections have shown how the additional hyperparameter $\alpha$ shapes GFlowNet training dynamics. We now evaluate whether these dynamics yield performance gains across various domains, particularly by enhancing discovery of distinct high-reward modes and preserving sample diversity.

### 5.1 DATASETS AND SETTINGS

We consider three popular benchmarks, Set Generation (Pan et al., 2023), non-autoregressive Bit Sequence Generation (Tiapkin et al., 2024) and Molecule Generation (Bengio et al., 2021). In contrast to former works, we separate training samples and evaluation samples in Bit Sequence Generation and Molecule Generation to avoid confusions in evaluation metrics, following the practice in Set Generation. We adopt a exponential annealing function in stage 2 of Alg. 1. Details of experimental settings are deferred to App. D.1.

**Set Generation.** The goal of this task is to generate sets of fixed sizes. The maximum capacity of sets $|S|$ and the size of the vocabulary vary from being small, medium to large, and task difficulty increases correspondingly. The sampling process of a set terminates when the number of elements reaches its maximum capacity. The reward function is defined as the accumulation of individual energy exponent $\exp(-\mathcal{E}(e_i))$ of each element $e_i$ in a set, i.e. $R(x) \triangleq \prod_{i=1}^{|S|} \exp(-\mathcal{E}(e_i))$, which equips FL variants with ideal local credits (Pan et al., 2023; Jang et al., 2024). Results are reported with 10 random seeds.

**Bit Sequence Generation.** The goal of this task is to generate binary strings of a fixed length $n = 120$. In each step, a k-bit word is sampled to replace an existing k-bit empty word. The sampling process

Table 1: Results on Set Generation. $\alpha$-GFNs perform better at reward and modes across all settings.

| Set Size | Metric | DB | | FL-DB | | TB | |
|---|---|---|---|---|---|---|---|
| | | Baseline | Ours | Baseline | Ours | Baseline | Ours |
| Small | Modes↑ | $26.6_{\pm 3.5}$ | $\mathbf{51.1}_{\pm 8.8}$ | $83.1_{\pm 1.9}$ | $\mathbf{88.5}_{\pm 2.7}$ | $23.2_{\pm 5.9}$ | $\mathbf{26.8}_{\pm 5.6}$ |
| | Top-1000 Reward↑ | $0.184_{\pm 0.001}$ | $\mathbf{0.203}_{\pm 0.004}$ | $0.218_{\pm 0.001}$ | $\mathbf{0.220}_{\pm 0.002}$ | $0.180_{\pm 0.001}$ | $\mathbf{0.184}_{\pm 0.001}$ |
| | Spearman Corr | $0.998_{\pm 0.001}$ | $0.995_{\pm 0.001}$ | $0.989_{\pm 0.004}$ | $0.976_{\pm 0.006}$ | $1.000_{\pm 0.000}$ | $1.000_{\pm 0.000}$ |
| Medium | Modes↑ | $0.0_{\pm 0.0}$ | $\mathbf{79.5}_{\pm 105.61}$ | $9.0_{\pm 5.57}$ | $\mathbf{255.6}_{\pm 249.8}$ | $0.0_{\pm 0.0}$ | $\mathbf{380.2}_{\pm 469.2}$ |
| | Top-1000 Reward↑ | $113916_{\pm 7869}$ | $\mathbf{575348}_{\pm 65238}$ | $498434_{\pm 26370}$ | $\mathbf{665591}_{\pm 42842}$ | $44847_{\pm 1539}$ | $\mathbf{668396}_{\pm 74759}$ |
| | Spearman Corr | $0.994_{\pm 0.002}$ | $0.977_{\pm 0.004}$ | $0.971_{\pm 0.007}$ | $0.946_{\pm 0.010}$ | $0.998_{\pm 0.000}$ | $0.906_{\pm 0.268}$ |
| Large | Modes↑ | $0.0_{\pm 0.0}$ | $\mathbf{215.0}_{\pm 183.1}$ | $223.9_{\pm 101.6}$ | $\mathbf{2475.4}_{\pm 721.8}$ | $0.0_{\pm 0.0}$ | $\mathbf{431.0}_{\pm 352.8}$ |
| | Top-1000 Reward↑ | $54440_{\pm 3964}$ | $\mathbf{580376}_{\pm 72592}$ | $600956_{\pm 31384}$ | $\mathbf{775196}_{\pm 18595}$ | $11318_{\pm 394}$ | $\mathbf{636234}_{\pm 63003}$ |
| | Spearman Corr | $0.979_{\pm 0.010}$ | $0.945_{\pm 0.010}$ | $0.886_{\pm 0.017}$ | $0.848_{\pm 0.020}$ | $0.996_{\pm 0.000}$ | $0.996_{\pm 0.001}$ |

terminates when there are $n/k$ non-empty words. The reward of a sample is defined as the negative exponent of its minimal Hamming distance to any target mode in a predefined set $M$ with $|M| = 60$, i.e. $R(x) = \exp(-\min_{y \in M} d(x, y))$. We vary $k \in \{2, 4, 6, 8, 10\}$ as Tiapkin et al. (2024). We consider DB, SubTB($\lambda$) and TB, and report results with 5 random seeds.

**Molecule Generation.** This task aims to design binders for the soluble epoxide hydrolase (sEH) protein by iteratively appending "blocks" from a fixed library onto a growing molecular graph (Jin et al., 2018). The reward is computed via a pretrained proxy from Bengio et al. (2021). We consider DB, FL-DB, SubTB($\lambda$), FL-SubTB($\lambda$) and TB, and report results with 5 random seeds.

## 5.2 RESULTS

**Enhanced Performance: More Modes and Reward.** As shown in Table 1, Table 2, and Table 3, the exploration–exploitation flexibility introduced by $\alpha$ improves both reward and the number of modes in most settings. Because modes correspond to distinct high-reward samples, their increase also indicates that sample diversity is preserved.

In **Set Generation**, results in Table 1 shows that $\alpha$-GFN objectives achieve 92%, 6%, 16% more number of modes for DB, FL-DB, TB respectively in small sets. Moreover, the performance gain scales well with increasing task difficulty. For medium and large sets, $\alpha$-GFN objectives achieve at least 1000% more modes than vanilla objectives. For TB and DB, the performance gain is even larger. The average reward also greatly increases, suggesting the benefit of enhanced reward exploration-exploitation abilities of $\alpha$-GFNs.

In **Bit Sequence Generation**, results in Table 2 still demonstrate the effectiveness of the hyperparameter $\alpha$. $\alpha$-GFNs find up to 8 more modes than vanilla GFlowNets on average and consistently surpass them in 13 settings, while vanilla GFlowNets can only find up to 2 modes than $\alpha$-GFNs in 2 settings.

Finally, we evaluate $\alpha$-GFNs on a real-world task, **Molecule Generation**. As shown in Table 3, $\alpha$-GFN objectives substantially outperform the baselines across most evaluation metrics. In terms of the number of modes, setting $\alpha \neq 0.5$ yields gains of 44% with DB, 177% with FL-DB, 19% with SubTB($\lambda$), 145% with FL-SubTB($\lambda$), and 5% with TB. Corresponding improvements are also observed in the reward metric, many of which are significant. Moreover, the sample similarity metrics are either improved or slightly increased, indicating that the diversity of high-reward samples is well preserved. Taken together, these results provide strong evidence for the effectiveness of $\alpha$-tuning.

Across experiments, we observe that $\alpha$-GFNs are comparable to vanilla GFlowNets in Spearman correlations between sampling probabilities and rewards. In reality, $\alpha$-GFNs even achieve a better fitting to the reward in

Table 2: Results on Bit Sequence Generation. In terms of number of modes on average, $\alpha$-GFN objectives outperform vanilla GFlowNet objectives across 87% task settings.

| k | Metric | DB | | SubTB($\lambda$) | | TB | |
|---|--------|----|----|------------------|----|----|----|
| | | Baseline | Ours | Baseline | Ours | Baseline | Ours |
| 2 | Modes | $0.60_{\pm 0.89}$ | $\mathbf{2.60}_{\pm 2.07}$ | $\mathbf{22.80}_{\pm 13.27}$ | $20.80_{\pm 11.39}$ | $12.20_{\pm 2.39}$ | $\mathbf{16.80}_{\pm 3.77}$ |
| | Spearman | $0.32_{\pm 0.43}$ | $0.50_{\pm 0.04}$ | $0.47_{\pm 0.06}$ | $0.51_{\pm 0.19}$ | $0.47_{\pm 0.17}$ | $0.54_{\pm 0.00}$ |
| 4 | Modes | $9.80_{\pm 6.50}$ | $\mathbf{12.00}_{\pm 1.58}$ | $35.40_{\pm 4.16}$ | $\mathbf{40.40}_{\pm 2.97}$ | $38.00_{\pm 2.74}$ | $\mathbf{41.80}_{\pm 5.07}$ |
| | Spearman | $0.58_{\pm 0.00}$ | $0.57_{\pm 0.00}$ | $0.58_{\pm 0.00}$ | $0.60_{\pm 0.02}$ | $0.58_{\pm 0.00}$ | $0.58_{\pm 0.00}$ |
| 6 | Modes | $4.20_{\pm 1.92}$ | $\mathbf{5.20}_{\pm 1.30}$ | $20.60_{\pm 3.36}$ | $\mathbf{22.20}_{\pm 7.33}$ | $21.60_{\pm 2.70}$ | $\mathbf{23.40}_{\pm 3.71}$ |
| | Spearman | $0.55_{\pm 0.01}$ | $0.56_{\pm 0.00}$ | $0.55_{\pm 0.00}$ | $0.59_{\pm 0.09}$ | $0.56_{\pm 0.00}$ | $0.56_{\pm 0.00}$ |
| 8 | Modes | $36.40_{\pm 1.82}$ | $\mathbf{44.40}_{\pm 3.78}$ | $58.60_{\pm 1.67}$ | $\mathbf{58.60}_{\pm 0.89}$ | $58.80_{\pm 1.30}$ | $\mathbf{59.00}_{\pm 0.00}$ |
| | Spearman | $0.79_{\pm 0.00}$ | $0.79_{\pm 0.00}$ | $0.81_{\pm 0.00}$ | $0.81_{\pm 0.00}$ | $0.81_{\pm 0.00}$ | $0.81_{\pm 0.00}$ |
| 10 | Modes | $6.80_{\pm 2.05}$ | $\mathbf{8.20}_{\pm 2.39}$ | $19.20_{\pm 4.92}$ | $\mathbf{21.60}_{\pm 3.05}$ | $\mathbf{24.40}_{\pm 3.36}$ | $23.80_{\pm 1.79}$ |
| | Spearman | $0.57_{\pm 0.01}$ | $0.57_{\pm 0.00}$ | $0.57_{\pm 0.00}$ | $0.57_{\pm 0.00}$ | $0.57_{\pm 0.00}$ | $0.57_{\pm 0.00}$ |

Table 3: Results on Molecule Generation. $\alpha$-GFNs are better at modes and reward for all objectives. Similarity metrics are improved or just slightly increased. Correlations of FL variants are omitted due to their biased target (Silva et al., 2025).

| Metric | DB | | FL-DB | | SubTB($\lambda$) | | FL-SubTB($\lambda$) | | TB | |
|--------|----|----|-------|----|------------------|----|---------------------|----|----|----|
| | Baseline | Ours | Baseline | Ours | Baseline | Ours | Baseline | Ours | Baseline | Ours |
| Modes↑ | $10.00_{\pm 3.08}$ | $\mathbf{14.40}_{\pm 2.41}$ | $9.00_{\pm 1.58}$ | $\mathbf{25.00}_{\pm 4.36}$ | $22.20_{\pm 2.77}$ | $\mathbf{26.40}_{\pm 2.30}$ | $16.00_{\pm 3.16}$ | $\mathbf{39.20}_{\pm 7.05}$ | $38.40_{\pm 6.11}$ | $\mathbf{40.20}_{\pm 5.07}$ |
| Top-10 Reward↑ | $7.16_{\pm 0.06}$ | $\mathbf{7.24}_{\pm 0.05}$ | $7.28_{\pm 0.06}$ | $\mathbf{7.37}_{\pm 0.07}$ | $7.30_{\pm 0.07}$ | $\mathbf{7.33}_{\pm 0.06}$ | $7.39_{\pm 0.08}$ | $\mathbf{7.46}_{\pm 0.05}$ | $\mathbf{7.45}_{\pm 0.06}$ | $7.45_{\pm 0.08}$ |
| Top-100 Reward↑ | $6.81_{\pm 0.03}$ | $\mathbf{6.84}_{\pm 0.03}$ | $6.97_{\pm 0.02}$ | $\mathbf{7.03}_{\pm 0.05}$ | $6.97_{\pm 0.03}$ | $\mathbf{6.99}_{\pm 0.02}$ | $6.98_{\pm 0.06}$ | $\mathbf{7.12}_{\pm 0.04}$ | $7.14_{\pm 0.05}$ | $\mathbf{7.16}_{\pm 0.05}$ |
| Top-1000 Reward↑ | $\mathbf{6.33}_{\pm 0.01}$ | $6.32_{\pm 0.01}$ | $\mathbf{6.69}_{\pm 0.06}$ | $6.47_{\pm 0.14}$ | $6.51_{\pm 0.01}$ | $\mathbf{6.52}_{\pm 0.01}$ | $6.48_{\pm 0.13}$ | $\mathbf{6.50}_{\pm 0.05}$ | $6.70_{\pm 0.07}$ | $\mathbf{6.73}_{\pm 0.07}$ |
| Top-10 Similarity↓ | $0.59_{\pm 0.01}$ | $\mathbf{0.55}_{\pm 0.05}$ | $0.59_{\pm 0.06}$ | $\mathbf{0.52}_{\pm 0.07}$ | $\mathbf{0.56}_{\pm 0.04}$ | $0.58_{\pm 0.02}$ | $0.55_{\pm 0.18}$ | $0.55_{\pm 0.18}$ | $\mathbf{0.62}_{\pm 0.03}$ | $0.66_{\pm 0.04}$ |
| Top-100 Similarity↓ | $0.57_{\pm 0.01}$ | $\mathbf{0.56}_{\pm 0.01}$ | $0.74_{\pm 0.05}$ | $\mathbf{0.51}_{\pm 0.03}$ | $\mathbf{0.57}_{\pm 0.01}$ | $0.58_{\pm 0.02}$ | $\mathbf{0.49}_{\pm 0.05}$ | $0.57_{\pm 0.01}$ | $\mathbf{0.63}_{\pm 0.03}$ | $0.64_{\pm 0.04}$ |
| Top-1000 Similarity↓ | $0.56_{\pm 0.00}$ | $0.56_{\pm 0.00}$ | $0.61_{\pm 0.03}$ | $\mathbf{0.50}_{\pm 0.05}$ | $0.56_{\pm 0.00}$ | $0.56_{\pm 0.01}$ | $0.54_{\pm 0.10}$ | $\mathbf{0.50}_{\pm 0.00}$ | $\mathbf{0.60}_{\pm 0.03}$ | $0.61_{\pm 0.04}$ |
| Spearman Corr | $0.53_{\pm 0.02}$ | $0.50_{\pm 0.05}$ | — | | $0.57_{\pm 0.07}$ | $0.59_{\pm 0.05}$ | — | | $0.22_{\pm 0.45}$ | $0.47_{\pm 0.16}$ |

many cases. In Molecule Generation, $\alpha$-TB delivers roughly $2\times$ improvement in Spearman correlations over vanilla TB with substantially lower variance across runs, and $\alpha$-SubTB($\lambda$) also performs better than vanilla SubTB($\lambda$) in this metric. When the correlation drops, $\alpha$-GFNs still consistently discover more modes. These results highlight the reward-fitting ability of $\alpha$-GFNs.

**Stability and Effects of Scheduling.** Although we observe clear performance gains across tasks, the standard deviation especially in the number of modes, sometimes increases, raising stability concerns. To probe this and the effect of annealing $\alpha$ on reward fitting, we conduct a case study on Molecule Generation. As shown in Fig. 4(a,b), $\alpha$-SubTB($\lambda$) consistently outperforms SubTB($\lambda$) in stage 1 (before annealing) with $\alpha = 0.9$ and maintains its advantage in stage 2 (annealing), where exponentially fast annealing induces mild loss oscillations that settle after roughly 8,000 steps, leaving a modest increase in mode variance. Nevertheless, a clear performance gain throughtout the training is still clearly observed. Meanwhile, Fig. 4(c) shows that annealing restores reward fitting. The Spearman correlation of $\alpha$-DB rebounds from around 0.4 before stage 2 to about 0.5 afterward. In summary, the two-stage schedule preserves the performance gains and recovers reward fitting at the cost of a brief, limited increase in variance in some cases.

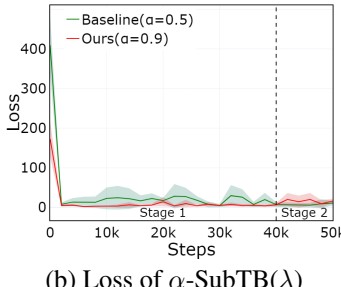
(a) Modes of $\alpha$-SubTB($\lambda$)

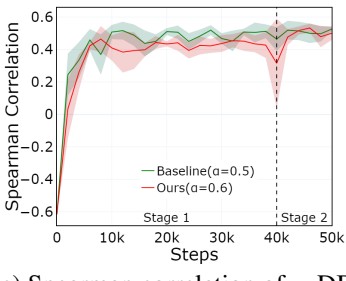
(b) Loss of $\alpha$-SubTB($\lambda$)

(c) Spearman correlation of $\alpha$-DB

Figure 4: A case study on **(a)** the mode curves of $\alpha$-FL-SubTB, **(b)** the loss curves of $\alpha$-FL-SubTB($\lambda$) and **(c)** the Spearman correlation curves of $\alpha$-DB in Molecule Generation during training.

**Length-Controlling Side Effects.** With a 'stop' action allowing $P_F$ to choose trajectory length, the trade-off in Prop. 4.5 induces an additional effect. As demonstrated in Fig. 5, as $\alpha$ is set larger, trajectories are lengthened since stronger exploitation may shift mass from 'stop' to constructive actions. Intriguingly, Fig. 5 also shows that average per-action entropy increases with trajectory length, plausibly reflecting accumulated uncertainty over longer horizons. A formal analysis is left to future work.

**Ablation Studies.** We assess the sensitivity of $\alpha$-GFNs via an ablation on the number of modes across $\alpha$ settings for molecule generation. As shown in Table 4, performance gains persist even when $\alpha$ is not tuned to its optimal value: for DB, FL-DB, SubTB($\lambda$), FL-SubTB($\lambda$), and TB, adjusting $\alpha$ generally yields more modes. These results indicate that $\alpha$-GFNs exhibit low sensitivity to precise $\alpha$ selection while still delivering consistent improvements.

Additional experimental results are in App. D.2.

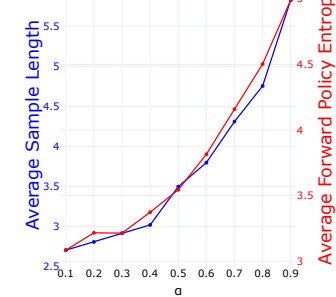

Figure 5: Sample length and entropy with $\alpha$-FL-SubTB in Molecule Generation.

Table 4: Number of modes vs. $\alpha$ in Molecule Generation. Entries surpassing the baseline are set in **boldface**.

| Modes↑ | $\alpha$ | | | | | | | | |
|---|---|---|---|---|---|---|---|---|---|
| **Objective** | 0.1 | 0.2 | 0.3 | 0.4 | 0.5 (Baseline) | 0.6 | 0.7 | 0.8 | 0.9 |
| DB | $5.40_{\pm0.89}$ | $10.00_{\pm4.18}$ | $\mathbf{11.20}_{\pm4.76}$ | $\mathbf{11.40}_{\pm3.21}$ | $10.00_{\pm3.08}$ | $\mathbf{14.40}_{\pm2.41}$ | $\mathbf{11.80}_{\pm3.96}$ | $\mathbf{11.00}_{\pm4.90}$ | $\mathbf{13.40}_{\pm3.21}$ |
| FL-DB | $5.80_{\pm1.48}$ | $5.40_{\pm1.14}$ | $6.40_{\pm3.13}$ | $\mathbf{9.17}_{\pm2.14}$ | $9.00_{\pm1.58}$ | $\mathbf{13.20}_{\pm4.66}$ | $\mathbf{14.60}_{\pm6.07}$ | $\mathbf{24.00}_{\pm3.61}$ | $\mathbf{25.00}_{\pm4.36}$ |
| SubTB($\lambda$) | $20.20_{\pm5.63}$ | $18.20_{\pm4.49}$ | $\mathbf{22.80}_{\pm3.77}$ | $\mathbf{24.00}_{\pm6.44}$ | $22.20_{\pm2.77}$ | $\mathbf{26.40}_{\pm2.30}$ | $22.00_{\pm5.43}$ | $\mathbf{25.20}_{\pm3.63}$ | $\mathbf{24.00}_{\pm5.79}$ |
| FL-SubTB($\lambda$) | $8.40_{\pm3.21}$ | $11.00_{\pm3.94}$ | $12.00_{\pm4.95}$ | $12.60_{\pm2.61}$ | $16.00_{\pm3.16}$ | $\mathbf{18.20}_{\pm5.31}$ | $\mathbf{23.40}_{\pm6.62}$ | $\mathbf{34.20}_{\pm5.67}$ | $\mathbf{39.20}_{\pm7.05}$ |
| TB | $19.80_{\pm3.83}$ | $29.60_{\pm7.30}$ | $37.00_{\pm7.42}$ | $34.40_{\pm6.54}$ | $38.40_{\pm6.11}$ | $\mathbf{40.20}_{\pm5.07}$ | $36.80_{\pm4.44}$ | $\mathbf{39.20}_{\pm9.81}$ | $\mathbf{60.60}_{\pm32.39}$ |

## 6 CONCLUSION

In this work, we uncover the implicit equal weighting of forward and backward policies in GFlowNet objectives through an extended connection to Markov chains. Building on this finding, we introduce $\alpha$-GFNs, which provide a controllable exploration–exploitation trade-off by mixing forward and backward policies with a hyperparameter $\alpha$. The convergence of these mixed objectives is established via their link to Markov chains. We further explain the role of different $\alpha$ values through a gradient-based analysis and propose a scheduled training algorithm that combines the benefits of multiple $\alpha$ values. Experiments across diverse

domains show that $\alpha$-GFNs consistently outperform vanilla GFlowNets by discovering more high-reward modes while maintaining sample diversity, highlighting the practical value of $\alpha$. These results strengthen the link between Markov chain theory and GFlowNet practice and open promising directions for future research.

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

# A  GFLOWNET OBJECTIVE DEFINITIONS AND RELATED DISCUSSIONS

## A.1  DEFINITIONS

In this section, we provide the formal definitions of vanilla GFlowNet training objectives where both the forward policy $P_F$ and the backward policy $P_B$ are used, including the Detailed Balance Loss (DB, Bengio et al. (2023)), Trajectory Balance Loss (TB, Malkin et al. (2022)) and the convex combination of SubTB, SubTB($\lambda$) (Madan et al., 2023) along with Forward-Looking (FL) (Pan et al., 2023) variants of DB (FL-DB) and SubTB($\lambda$) (FL-SubTB($\lambda$)).

**Definition A.1.** (DB, Bengio et al. (2023)) Given a state flow function $F(\cdot) : S \to \mathbb{R}_+$, a forward policy $P_F(\cdot|\cdot) : \mathbb{A} \to [0, 1]$, a backward policy $P_B(\cdot|\cdot) : \mathbb{A} \to [0, 1]$, Detailed Balance targets at

$$F(s)P_F(s'|s) = F(s)P_B(s|s'), \quad \text{for every } (s, s') \in \mathbb{A} \tag{5}$$

where $\forall s \in \mathcal{X}, F(s)P_F(s_f|s) = R(s)$. And the corresponding loss function is

$$L_{\text{DB}}(s, s') = \log^2 \left( \frac{F(s)P_F(s'|s)}{F(s')P_B(s|s')} \right). \tag{6}$$

**Definition A.2.** (TB, Malkin et al. (2022)) Given a state flow function $F(\cdot) : S \to \mathbb{R}_+$, a forward policy $P_F(\cdot|\cdot) : \mathbb{A} \to [0, 1]$, a backward policy $P_B(\cdot|\cdot) : \mathbb{A} \to [0, 1]$, for a complete trajectory $\mathfrak{t}^f = (s_0 = s_s, s_1, \ldots, s_{n-1} = x, s_n = s_f) \in \mathfrak{T}$ where $\mathfrak{T}^{\text{flow}}$ is the set of complete trajectories in the graph, TB targets at

$$F(s_0)P_F(s_f|x) \prod_{i=1}^{n-1} P_F(s_i|s_{i-1}) = R(s_n) \prod_{i=1}^{n-1} P_B(s_{i-1}|s_i) \tag{7}$$

And the corresponding loss function is

$$L_{\text{TB}}(\mathfrak{t}^f) = \log^2 \left( \frac{F(s_0)P_F(s_f|x) \prod_{i=1}^{n-1} P_F(s_i|s_{i-1})}{R(s_n) \prod_{i=1}^{n-1} P_B(s_{i-1}|s_i)} \right). \tag{8}$$

Note that $F(s_0) = F(s_f) = \sum_{x \in \mathcal{X}} R(x)$ and $P_B(x|s_f) = \frac{R(x)}{\sum_{x \in \mathcal{X}} R(x)}$.

**Definition A.3.** (SubTB, Madan et al. (2023)) Given a partial trajectory $\mathfrak{t}' = (s_k, s_{k+1}, \ldots, s_{k+m}) \subset \mathfrak{t}^f \in \mathfrak{T}^{\text{flow}}$, a state flow function $F(\cdot) : S \to \mathbb{R}_+$, a forward policy $P_F(\cdot|\cdot) : \mathbb{A} \to [0, 1]$, a backward policy $P_B(\cdot|\cdot) : \mathbb{A} \to [0, 1]$, SubTB targets at Eq. 1, and the corresponding loss function is

$$L_{\text{SubTB}}(\mathfrak{t}') = \log^2 \left( \frac{F(s_k) \prod_{i=1}^m P_F(s_{k+i}|s_{k+i-1})}{F(s_{k+m}) \prod_{i=1}^m P_B(s_{k+i-1}|s_{k+i})} \right). \tag{9}$$

**Definition A.4.** (SubTB($\lambda$), Madan et al. (2023)) Given a complete trajectory $\mathfrak{t}^f = (s_0 = s_s, s_1, \ldots, s_{n-1} = x, s_n = s_f) \in \mathfrak{T}^{\text{flow}}$, define an extracted subtrajectory $\mathfrak{t}_{i:j}$ to be

$$\mathfrak{t}_{i:j} = (s_i, s_{i+1}, \ldots, s_j), \quad 0 \le i < j \le n. \tag{10}$$

Then, SubTB($\lambda$) loss for the complete trajectory $\mathfrak{t}$ is a convex combination of SubTB loss at these partial trajectories, i.e.

$$L_{\text{SubTB}}(\mathfrak{t}^f, \lambda) = \frac{\sum_{0 \le i < j \le n} \lambda^{j-i} L_{\text{SubTB}}(\mathfrak{t}_{i:j})}{\sum_{0 \le i < j \le n} \lambda^{j-i}} \tag{11}$$

where $\lambda \in (0, +\infty)$.

Additionally, we consider the Forward Looking (FL) (Pan et al., 2023) variants of DB and SubTB($\lambda$), termed FL-DB and FL-SubTB($\lambda$) respectively. We start with an introduction to the state-level and edge-level energy function.

**Definition A.5.** (Pan et al., 2023) Given a state-level energy function $\mathcal{E}(\cdot) : S \to \mathbb{R}$, the edge-level energy function is an extension of its state-level counterpart, i.e.

$$\mathcal{E}(s, s') = \mathcal{E}(s') - \mathcal{E}(s) \quad \text{for all } (s, s') \in \mathbb{A}. \tag{12}$$

Applying Def. A.5 to DB and SubTB($\lambda$), the FL variants are obtained:

**Definition A.6.** (FL-DB, Pan et al. (2023)) Based on the the definition of DB in Def. A.1, with an edge-level energy function $\mathcal{E}(\cdot, \cdot) : \mathbb{A} \to \mathbb{R}$, Forward-Looking Detailed Balance (FL-DB) targets at:

$$F(s)P_F(s'|s) = F(s')P_B(s|s')e^{-\mathcal{E}(s,s')} \tag{13}$$

and the corresponding loss function is

$$L_{\text{FL-DB}}(s, s') = \log^2 \left( \frac{F(s)P_F(s'|s)}{F(s')P_B(s|s')e^{-\mathcal{E}(s,s')}} \right). \tag{14}$$

**Definition A.7.** (FL-SubTB($\lambda$), Pan et al. (2023)) Based on the definition of SubTB in Def. A.3, given a partial trajectory $\mathfrak{t}_{i:j} = (s_i, s_{i+1}, \ldots, s_j) \subset \mathfrak{t} \in \mathfrak{T}^{\text{flow}}$ and an edge-level energy function $\mathcal{E}(\cdot, \cdot) : \mathbb{A} \to \mathbb{R}$, FL-SubTB targets at

$$F(s_k) \prod_{i=1}^{m} P_F(s_{k+i} \mid s_{k+i-1}) \ = \ F(s_{k+m}) \prod_{i=1}^{m} P_B(s_{k+i-1} \mid s_{k+i}) e^{-\mathcal{E}(s_{k+i-1}, s_{k+i})}.$$

and the corresponding loss function is

$$L_{\text{FL-SubTB}}(\mathfrak{t}_{i:j}) = \log^2 \left( \frac{F(s_i) \prod_{k=1}^{j} P_F(s_{i+k}|s_{i+k-1})}{F(s_j) \prod_{k=1}^{j} P_B(s_{i+k-1}|s_{i+k})} \right) \tag{15}$$

Following Def. A.4, FL-SubTB($\lambda$) is a convex combination of Fl-SubTB. Given $\lambda \in (0, +\infty)$, FL-SubTB($\lambda$) is

$$L_{\text{FL-SubTB}}(\mathfrak{t}^f, \lambda) = \frac{\sum\limits_{0 \le i < j \le n} \lambda^{j-i} L_{\text{FL-SubTB}}(\mathfrak{t}_{i:j})}{\sum\limits_{0 \le i < j \le n} \lambda^{j-i}}. \tag{16}$$

## A.2 Discussions

To further demonstrate that $\alpha$-GFNs are compatible with vanilla GFlowNet training techniques, we formalize FL (Pan et al., 2023) within the Markov chain framework in a matrix form, which suggests FL is a prior to the probability measures:

**Theorem A.8.** *FL adds a prior to the probability measures of a GFlowNet, i.e.* $\pi = \tilde{\pi}\mathcal{E}$, *where* $\mathcal{E} = diag(e^{-\mathcal{E}(s_0)}, \ldots, e^{-\mathcal{E}(s_f)})$ *is the diagonal matrix constructed with exponent of the energy function* $\mathcal{E}(\cdot)$.

*Proof.* Assumption 4.1 and Prop. 4.2 in Pan et al. (2023) suggests

$$F = \tilde{F}\mathcal{E}$$

is the FL reparameterization of flows, where $F = (F(s_s), \ldots, F(s_f))$ is the vector of flows, and $\tilde{F} = (\tilde{F}(s_s), \ldots, \tilde{F}(s_f))$ is the reparameterized counterpart. Coupling with the fact that state flows are state-level probability measures (Deleu & Bengio, 2023), it is direct that

$$\pi = \tilde{\pi}\mathcal{E}$$

where $\pi$ and $\tilde{\pi}$ are the probability measures corresponding to $F$ and $\tilde{F}$ respectively. $\qquad\square$

## B Supporting Theoretical Framework

The following theoretical part addresses the fundamental results of our paper under the assumption that the state space is finite. This assumption is reasonable since the state space of GFlowNets are compositional (Bengio et al., 2021; 2023). Throughout, we denote by $P$ the true transition probability of the graph to be learned, which corresponds to the forward policy $P_F$ in GFlowNets. We first summarize the results we prove and then prove the results one by one in subsections.

(i) **From GFN to MC**: We show that GFlowNets can be represented as irreducible Markov Chains.

   In particular, the resulting Markov chain is positive recurrent, admits a unique stationary distribution $\pi$ to which the Markov chain converges from any state.

(ii) **From MC to GFN under constraints**: We show that every reducible Markov chain with transition probability $P$ satisfying a specific finite set of linear constraints is a GFlowNet.

   We call such Markov chains **GFNMC** (GFlowNet Markov chain). Such a Markov chain allows for a detailed balance $\tilde{P}$, which is again a GFNMC sharing the same trajectories (in the reverse direction), the same stationary distribution as well as the same eigenvalues (not necessarily the same eigenvectors).

   Furthermore any $\alpha$-GFNMC is again irreducible, share the same stationary distribution but not necessarily the same eigenvalues.

(iii) **Convergence rate**: We show that the periodicity of every GFNMC (GFlowNet Markov Chain) is equal to the greatest common denominator of all the trajectory length. It is in particular the case for any reasonable graph with various trajectory length to be aperiodic (periodicity of 1).

   If not, for instance if all trajectories share the same length, we show that for any $0 < \alpha < 1$, the periodicity of the $\alpha$-GFN is either 2 or 1.

   This is important in terms of convergence as aperiodic Markov chains are ergodic geometric, that is the convergence to the stationary distribution also holds in total variation as exponential rate determined by the second largest eigenvalue. In terms of sampling, such results allow for a central limit theorem akin to classical Monte Carlo.

   Denote with $\beta$, $\tilde{\beta}$ and $\beta_\alpha$ the second largest eigenvalues. While $\beta = \tilde{\beta}$, there is no result as $\beta_\alpha$ as a function of $\beta$ as it is highly non-linear. Nevertheless, it might drastically improve by mixing. It justifies the approach to take the hyper parameter $\alpha$ as trainable too to achieve a good convergence and a good sampling result.

(iv) **Freedom of the Choice of the Backward Policy**: So far, the analysis of $\alpha$-GFN has been done by considering $P_\alpha = \alpha P + (1 - \alpha)\tilde{P}$ where $\tilde{P}$ is exactly the time reversal of $P$. However, in practice, the training to learn $P$ is done by using a different, usually fixed backward policy $P_B$ that is proportional to the reward in the last step of the graph. A loss function furthermore enforces some weak to strong form of reversibility. It is apriori not clear that the resulting stationary distribution will be the one of $P$.

   We, however, show that after training $P_\alpha$ will share the same stationary distribution as $P$ and $P_B$. Furthermore, if after training $P_\alpha$ set the different losses taken into account to 0, then $P_\alpha$ would satisfy the corresponding balanced constraints.

Prior work (Deleu & Bengio, 2023) touches on **(i)**, but it does not fully formalize the Markov-chain-to-GFlowNet (MC→GFN) connection. In particular, key Markov chain details underlying this link are left unspecified, and the remaining aspects, **(ii)**, **(iii)**, and **(iv)**, have not, to our knowledge, been investigated. We address these gaps and develop a more complete account of the relationship between GFlowNets and Markov chains.

### B.1 GFNs ARE MCs

We start with giving a detailed version of notations in Sec. 2. Following Bengio et al. (2021), we consider a *directed graph* $(S, \mathbb{A})$ where $S$ is a finite state space and $\mathbb{A} \subseteq S \times S$ is a set of *edges* between states where $s \to s' = (s, s') \in \mathbb{A}$ denotes an edge. A *trajectory* is a finite sequence $\mathfrak{t}^f = (s_0, \ldots, s_N)$ where $s_n \to s_{n+1}$ is an edge for any $n \leq N - 1$ where $N \geq 1$. The directed graph is called *acyclic* if every such trajectory satisfies $s_0 \neq s_N$. A directed acyclic graph is called *pointed* if there exists a *source state* $s_s$ and a *final state* $s_f$ such that for every other state $s$, there exists a trajectory starting from $s_s$, running through $s$ and ending in $s_f$. Any such finite trajectory starting in $s_s$ and ending in $s_f$ is called *complete*.[2] From now one we only consider pointed directed acyclic graphs where the GFlowNet theoretical framework is built (Bengio et al., 2023) and denote by $\mathfrak{T}^{\mathrm{flow}}$ the set of complete trajectories $\mathfrak{t}^f = (s_0, \ldots, s_N)$ where $s_0 = s_s$ and $s_N = s_f$.

To realise the link with Markov chains, let $\mathfrak{T} = S^{\mathbb{N}_0}$ be the set of infinite trajectories $\mathfrak{t} = (s_0, s_1, \ldots)$ endowed with the product $\sigma$-algebra $\mathcal{T} = \otimes S$ where $S = 2^S$. We further denote by $X = (X_t)$ the *canonical process* on $\mathfrak{T}$ where $X_n(\mathfrak{t}) = s_n$ is the $n$-th state of the trajectory $\mathfrak{t}$. In other terms $X$ is the identity on $\mathfrak{T}$ since $X(\mathfrak{t}) = \mathfrak{t}$.[3] We finally denote by $\mathcal{T}_n = \sigma(X_m \colon m \leq n)$ the filtration generated by the canonical process.

Using a finite Kolmogorov extension argument, Bengio et al. (2021) show that any Markovian probability measure $\mathbf{P}^{\mathrm{flow}}$ on $\mathfrak{T}^{\mathrm{flow}}$ is uniquely given by a transition probability $P(s'|s) = P_{ss'}$ for every edge $s \to s'$.

*Remark* B.1. Throughout, we assume that $P(s'|s) > 0$ for every edge $s \to s'$.

This transition probability can be extended to $S \times S$ by defining $P(s_s|s_f) = 1$ and $P(s'|s) = 0$ for any other pair $s \to s'$ which is not an edge. In other terms the probability of returning to the source state from the final state is equal to 1. Such an extension also defines a Markovian probability $\mathbf{P}^{s_s}$ on the space of infinite trajectories with corresponding Markov chain starting at the source state $s_s$. The following theorem embed Markovian GFlowNets into a subset of Markov Chains with specific properties.

**Theorem B.2.** *The Markovian GFlowNet probability $\mathbf{P}^{flow}$ **coincides exactly** with $\mathbf{P}^{s_s}$ on the set of complete trajectories $\mathfrak{T}^{flow}$. Furthermore, the resulting Markov Chain is **irreducible**.*

In order to state Thm. B.2 correctly as $\mathfrak{T}^{\mathrm{flow}}$ is not even a subset of $\mathfrak{T}$, let us consider random times and stopping times.

**Definition B.3.** A function $\tau \colon \mathfrak{T} \to \mathbb{N}_0 \cup \{\infty\}$ is called a *random time* if $\tau$ is measurable and a *stopping time* if $\{\tau \leq n\}$ is in $\mathcal{T}_n$ for every $n$.

Typical example of stopping times are the first time a trajectory $\mathfrak{t}$ satisfies a condition. In particular, we make use of
$$\tau^s(\mathfrak{t}) = \inf\{n \colon X_n(\mathfrak{t}) = s\} \quad \text{and} \quad \tau^s_+(\mathfrak{t}) = \inf\{n \geq 1 \colon X_n(\mathfrak{t}) = s, n > 0\},$$
which are the first time and first time $\geq 1$ such that the trajectory $\mathfrak{t}$ visits the state $s$, respectively. Given a stopping time $\tau < \infty$, we denote by $X_\tau(\mathfrak{t}) = \mathfrak{t}_{\tau(\mathfrak{t})}$ the value of the trajectory at this particular random time $\tau(\mathfrak{t})$, as well as the stopped Markov chain $X^\tau := (X_{n \wedge \tau})$. The stopped Markov chain is just a slice of the trajectory $\mathfrak{t}$ between 0 and $\tau(\mathfrak{t})$ and constant afterwards:
$$X^\tau(\mathfrak{t}) = (\underbrace{s_0, \ldots, s_{\tau(\mathfrak{t})}}_{\text{slice before } \tau(\mathfrak{t})}, \underbrace{s_{\tau(\mathfrak{t})}, \ldots}_{\text{constant after}}).$$

Going back to flows, we define
$$\sigma(\mathfrak{t}) = \begin{cases} N & \text{if } (s_0, \ldots, s_N) \in \mathfrak{T}^{\mathrm{flow}} \text{ is a complete flow trajectory for some } N \\ 0 & \text{otherwise} \end{cases}$$

---

[2]Note that by definition, any state in $S$ is element of a complete trajectory.

[3]Note that any Markov chain is about defining a probability measure on the space of trajectories $\mathfrak{T}$ making the canonical process to satisfy the Markov property.

as the function returning the length of the complete flow trajectory if the infinite trajectory starts in $\mathfrak{T}^{\text{flow}}$ and $0$ otherwise.

**Lemma B.4.** *The function $\sigma$ is a uniformly bounded random time. Furthermore, $\mathfrak{T}^{\text{flow}}$ is in bijection with $\{\mathfrak{t}\colon \sigma(\mathfrak{t}) \geq 1\}$ which is a measurable subset of $\mathfrak{T}$.*

*Proof.* Since the state space is finite, it follows that $\sigma < \#S + 1$ and therefore uniformly bounded. It follows that the number of elementary flows is finite and therefore $\sigma$ is a finite sum of simple random variables greater than $1$, hence a random variable. Finally, by definition $\mathfrak{T}^{\text{flow}}$ is in bijection with $\{\sigma \geq 1\}$ and therefore measurable since $\sigma$ is measurable. $\qquad\square$

*Remark* B.5. If the state space is infinite, then elementary flows might be of arbitrary length. Hence, the set of elementary flows might be of uncountable cardinality. In this case, the measurability argument does not hold without further assumptions. Also, even if $\sigma$ is measurable, it is not clear a priori that it is a stopping time, and in general likely not.

Even if $\sigma$ is not a stopping time, it coincides with the stopping time $\tau^{s_f}$ with probability $1$ for the Markov probability starting from $s_s$.

**Proposition B.6.** *Let $\mathbf{P}^{s_s}$ be the Markovian probability starting from the source state $s_s$. Then it follows that*

$$\mathbf{P}^{s_s}[\sigma = \tau^{s_f}] = 1$$

*and $\mathbf{P}^{s_s}$ coincides with $\mathbf{P}^{\text{flow}}$ in the sense that for any $\mathfrak{t}^f$ in $\mathfrak{T}^{\text{flow}}$ it holds*

$$\mathbf{P}^{s_s}[X_{[0:\tau^{s_f}]} = \mathfrak{t}^f] = \mathbf{P}^{\text{flow}}[\mathfrak{t}^f].$$

*Proof.* It is clear that for any trajectory $\mathfrak{t}$ such that $\sigma(\mathfrak{t}) \geq 1$, it holds that $\tau^{s_f}(\mathfrak{t}) = \sigma(\mathfrak{t})$. And by the definition of $\mathbf{P}^{s_s}$ from the same transition probability, we get that

$$\mathbf{P}^{s_s}[X_{[0:\tau^{s_f}]} = \mathfrak{t}^f] = \mathbf{P}^{s_s}[X_{[0:\sigma]} = \mathfrak{t}^f] = \mathbf{P}^{\text{flow}}[\mathfrak{t}^f]$$

In particular, $\mathbf{P}^{s_s}$ has measure $1$ on the set where $\{\mathfrak{t}\colon \sigma(\mathfrak{t}) \geq 1\}$. $\qquad\square$

Recall that a Markov chain is called **irreducible** if for every pair of states $s, s'$ it holds that $\mathbf{P}^s[\tau^{s'} < \infty] = 1$ for any two states $s$ and $s'$.

**Proposition B.7.** *The Markov chain resulting from $P$ is irreducible.*

*Proof.* Any flow trajectory in $\{\sigma \geq 1\}$ starting in $s_s$ will reach $s_f$ with probability $1$ in a bounded amount of steps. Furthermore, from $P(s'|s) > 0$ for any edge $s \to s'$ shows that each state is visited by a flow trajectory with strict positive probability. The assumption that $P(s_s|s_f) = 1$ and the fact that $P(s_s|s_s) = 0$ shows that starting from any state $s$, the probability to reach $s_f$ and therefore $s_s$ is equal to $1$. From $s_s$ to any other state $s' \neq s_s$ is with strict positive probability, hence from strong Markov property, it follows that any state $s'$ is accessible from any state $s$ with strict positive probability, that is $\mathbf{P}^s[\tau^{s'} < \infty] = 1$. $\qquad\square$

Since the Markov chain with respect to the transition kernel $P$ is defined over a finite discrete state space, it is also **Harris recurrent** and **positive recurrent**.

## B.2 MC Constraints to be a GFN

*Remark* B.8. For an easy definition of pointed directed graphs, it is better to distinguish between source state $s_s$ and final state $s_f$. However, since the transition from $s_s$ to $s_f$ is with probability one, from Markov Chain perspective, it is equivalent to identify them both by setting $s_s = s_f = \bar{s}$, which is also the practice in Deleu & Bengio (2023).

We consider a transition probability $P$ such that the resulting Markov chain is irreducible. In particular, it is positive recurrent and Harris recurrent and has a stationary distribution $\pi$ since it is defined over a finite discrete state space. For any state $s$, $\tau^s$ as well as $\tau_+^s$ are finite stopping time with finite expectation.

For this Markov Chain to be a GFlowNet, it is necessary and sufficient that any finite trajectory $t^f = (s_0 = \bar{s}, s_1, \ldots, s_N = \bar{s})$ does not contain inner loop. In other terms for any state $s \neq \bar{s}$, the probability of returning to $s$ is 1 and it shall pass first through $\bar{s}$ also with probability 1.

**Theorem B.9.** *A transition probability $P$ for an irreducible Markov chain is a GFlowNet if and only if*

$$\mathbf{P}^s[\tau^{\bar{s}} < \tau_+^s] = 1 \quad \text{for every } s \neq \bar{s}. \tag{17}$$

*The condition in* Eq. 17 *translates into*

$$\pi_s(Z_{\bar{s}\bar{s}} - Z_{s\bar{s}}) + \pi_{\bar{s}}(Z_{ss} - Z_{\bar{s}s}) = \pi_{\bar{s}},$$

*where $Z$ is the fundamental matrix*

$$Z := (I - P + \Pi)^{-1} = \sum (P - \Pi)^n$$

*with $\Pi$ being the matrix with each row equal to $\pi$.*

Here, we use $Z$ to align with the classical Markov chain theory. **Note that this $Z$ is not the amount of flows in GFlowNets.**

*Proof.* The first statement Eq. 17 is clearly equivalent to the Markov Chain is concentrated onto a set of complete trajectories that corresponds to a pointed acyclic directed graph. In particular $\tau_+^{\bar{s}} \leq \#S$. Now from (Aldous & Fill, 2002, Corollary 2.8), it holds that

$$\mathbf{P}^s[\tau^{\bar{s}} < \tau_+^s] = \frac{1}{\pi_s(E^s[\tau^{\bar{s}}] + E^{\bar{s}}[\tau^s])}$$

while (Aldous & Fill, 2002, Lemma 2.12) states that

$$\pi_s E^s[\tau^{\bar{s}}] = \frac{\pi_s}{\pi_{\bar{s}}}(Z_{\bar{s}\bar{s}} - Z_{s\bar{s}}) \quad \text{and} \quad \pi_s E^{\bar{s}}[\tau^s] = (Z_{ss} - Z_{\bar{s}s})$$

Together with Eq. 17, it yields

$$\pi_s(Z_{\bar{s}\bar{s}} - Z_{s\bar{s}}) + \pi_{\bar{s}}(Z_{ss} - Z_{\bar{s}s}) = \pi_{\bar{s}}, \quad \text{for all } s \neq \bar{s}.$$

$\square$

Given an irreducible Markov chain with transition probability $P$ and resulting stationary distribution $\pi$ we can define the balanced chain $\tilde{P}$ as

$$\tilde{P}_{ss'} = \frac{\pi_{s'}}{\pi_s} P_{s's}$$

**Proposition B.10.** *The Markov chain $\tilde{P}$ is again irreducible with same stationary distribution $\pi$ and eigenvalues as $P$. Furthermore, it is also a GFNMC.*

*Proof.* The first statement is classical. Let us show that $\tilde{P}$ is a GFNMC. Denoting with $D = \mathrm{diag}(\pi)$, it holds that $\tilde{P} = D^{-1}P^\top D$. It is easy to check that $D\Pi D^{-1} = \Pi^\top$. It follows that

$$Z = (I - \tilde{P} - \Pi)^{-1} = (I - D^{-1}P^\top D - D^{-1}\Pi^\top D)^{-1}$$

$$= D^{-1}\left((I - P - \Pi)^\top\right)^{-1} D$$

$$= D^{-1}\left((I - P - \Pi)^{-1}\right)^\top D$$

$$= D^{-1}Z^\top D$$

showing that the fundamental matrices satisfy the same balance equation. It follows that

$$\pi_s\left(\tilde{Z}_{\bar{s}\bar{s}} - \tilde{Z}_{s\bar{s}}\right) + \pi_{\bar{s}}\left(\tilde{Z}_{ss} - \tilde{Z}_{\bar{s}s}\right) = \pi_s\left(Z_{\bar{s}\bar{s}} - \frac{\pi_{\bar{s}}}{\pi_s}Z_{\bar{s}s}\right) + \pi_{\bar{s}}\left(Z_{ss} - \frac{\pi_s}{\pi_{\bar{s}}}Z_{s\bar{s}}\right)$$

$$= \pi_s\left(Z_{\bar{s}\bar{s}} - Z_{s\bar{s}}\right) + \pi_{\bar{s}}\left(Z_{ss} - Z_{\bar{s}s}\right)$$

$$= \pi_{\bar{s}}$$

showing that $\tilde{P}$ is a GFNMC according to Thm. B.9. $\qquad\square$

Taking $0 < \alpha < 1$, the $\alpha$-GFN given by $\alpha P + (1 - \alpha)\tilde{P}$ is clearly irreducible with the same stationary distribution. However, it is no longer of GFNMC type as some inner loop might be present in trajectories from $\bar{s}$ to $\bar{s}$. It also do not share the same set of eigenvalues. Nevertheless, it is closely related to the properties of $\alpha$-GFNs, as demonstrated in the following.

### B.3 Convergence Rate

Let $P$ be the transition probability of an irreducible Markov chain of GFNMC type. Let $\pi$ be the stationary distribution, $\tilde{P}$ the balanced chain. We further denote by $\beta$ the largest modulus of the eigenvalue of $P$ which is not 1 and $\beta_\alpha$ the largest modulus of the eigenvalue of $P_\alpha = \alpha P + (1 - \alpha)\tilde{P}$. Let further $d(s) = \gcd\{n \colon P_{ss}^n > 0\}$ be the periodicity of the Markov chain. It is a classical result that for irreducible chains, this periodicity is the same for each state. Hence, let $d$, $\tilde{d}$ and $d_\alpha$ be the corresponding periodicities of $P$, $\tilde{P}$ and $\alpha P$, respectively.

**Proposition B.11.** *It holds that the periodicity of $P$ is equal to the greatest common divisor of the length of all complete trajectories. Furthermore $d = \tilde{d}$.*

*Finally for $0 < \alpha < 1$, it holds that*

$$d_\alpha = \begin{cases} 1 & \text{if } d \text{ is odd} \\ 1 \text{ or } 2 & \text{if } d \text{ is even} \end{cases}$$

*Proof.* It is clear that the periodicity of $P$ and therefore $\tilde{P}$ will coincide with the greatest common divisor of all trajectory lengths.

However for $0 < \alpha < 1$, it is then possible to have trajectories $s \to s' \to s$ with strict positive probability (since it can go with strict probability from $s$ to $s'$ with $P$ and strict positive probability from $s'$ to $s$ with $\tilde{P}$). Hence the periodicity of $\alpha P + (1 - \alpha)\tilde{P}$ divides $d$ as well as 2 showing the result. $\qquad\square$

*Remark* B.12. In the case of GFN, usually with very large graphs, either you have many possible lengths that make the periodicity already 1, or all trajectories are of the same length, and the periodicity is either 1 or 2 for the $\alpha$-GFN. Although $P$ in this case have a very large periodicity, $\tilde{P}$ will drastically reduce it to at least 2 if not 1.

Having a periodicity of $1$ is important in terms of convergence, as it is geometric ergodic. In this case the convergence to the stationary distribution also holds in total variation as exponential rate determined by the second largest eigenvalue and a central limit theorem also holds that ensure correct bounds in terms of sampling akin to Monte Carlo. The rate of convergence is dominated by the second largest eigenvalue. However, as eigenvalues are highly non-linear, it is not possible to find a direct relation between $\beta_\alpha$ and $\beta$. Nevertheless, it is highly possible that mixing with different $\alpha$ can improve the convergence rate depending on the structure of the original Markov Chain. **In particular, setting $\alpha = 0.5$ directly yields the discussions corresponding to vanilla GFlowNets.**

## C    PROOFS

App. B enables formal discussions on GFlowNets from the Markov chain perspective. Building on this aspect, we now take a closer look into the objectives of $\alpha$-GFNs and vanilla GFlowNets. Before diving into the corresponding theorems and propositions in the main body, we first show an equivalence between flows and probability measures with a generalization of Deleu & Bengio (2023). Although this equivalence has been used in previous parts following Deleu & Bengio (2023), we give a generalization for clarity in the following.

By defining

$$Z_{\text{state}} = \sum_{s \in S} F(s), \quad \pi(s) = \frac{F(s)}{Z_{\text{state}}}, \tag{18}$$

we directly obtain the probability measures at each state $s \in S$. Note that $\pi(\cdot)$ is a probability measure since $\sum_{s \in S} \pi(s) = 1$.

Equipped with Eq. 18 and App. B, we now derive the proofs for corresponding statements.

### C.1    PROOF FOR PROP. 4.1

Since $P_{0.5} = \frac{1}{2}P_F + \frac{1}{2}P_B$, given the partial trajectory $\mathfrak{t}' = (s_k, s_{k+1}, \ldots, s_{k+m})$, the reversibility of $P_{0.5}$ suggests

$$\pi(s_k) \prod_{i=1}^{m} P_{0.5}(s_{k+i}|s_{k+i-1}) = \pi(s_{k+m}) \prod_{i=1}^{m} P_{0.5}(s_{k+i-1}|s_{k+i})$$

which extends to

$$\pi(s_k) \prod_{i=1}^{m} \left( \frac{1}{2}P_F(s_{k+i}|s_{k+i-1}) + \frac{1}{2}P_B(s_{k+i}|s_{k+i-1}) \right) = \pi(s_{k+m}) \prod_{i=1}^{m} \left( \frac{1}{2}P_F(s_{k+i-1}|s_{k+i}) + \frac{1}{2}P_B(s_{k+i-1}|s_{k+i}) \right)$$

By noticing the fact that both $P_F$ and $P_B$ are one-directional within this partial trajectory, i.e. $\forall (s, s') \subset \mathfrak{t}'$, if $P_F(s'|s) > 0$, then $P_F(s|s') = 0, P_B(s|s') > 0, P_B(s'|s) = 0$, it follows that

$$(\frac{1}{2})^m \pi(s_k) \prod_{i=1}^{m} P_F(s_{k+i}|s_{k+i-1}) = (\frac{1}{2})^m \pi(s_{k+m}) \prod_{i=1}^{m} P_B(s_{k+i-1}|s_{k+i})$$

By eliminating $(\frac{1}{2})^m$ on both sides of the equation, one has

$$\pi(s_k) \prod_{i=1}^{m} P_F(s_{k+i}|s_{k+i-1}) = \pi(s_{k+m}) \prod_{i=1}^{m} P_B(s_{k+i-1}|s_{k+i})$$

### C.2    PROOF FOR THM. 4.2

**We first prove the claim that DB, SubTB and TB correspond to edge-level, partial-trajectory-level and trajectory-level reversibility of $P_{0.5}$.** Using Eq. 18, we obtain that for DB,

$$\frac{F(s)}{Z_{\text{state}}} P_F(s'|s) = \frac{F(s)}{Z_{\text{state}}} P_B(s|s')$$

which translates into

$$\pi(s) P_F(s'|s) = \pi(s') P_B(s|s').$$

Obviously, this equation is the reversibility in Prop. 4.1 at an edge-level. Similarly, one can obtain the proofs for SubTB and TB by plugging Eq. 18 into Eq. 1 and Eq. 7 and comparing the equations with Prop. 4.1.

**Next, we show that such objectives converge to unique state flows from the Markov chain perspective.**

For **DB** and **SubTB**($\lambda$), they both account for the edge-level reversibility (see Def. A.1 and Def. A.4), whereas reaching the edge-level reversibility is actually equivalent to the convergence of DB and, particularly, SubTB($\lambda$). The corresponding proof for DB is straightforward since the edge-level reversibility is identically its training target (see Def. A.1). For SubTB($\lambda$), by noticing that given a complete trajectory $t^f = (s_0 = s_s, s_1, \ldots, s_{n-1} = x, s_{n+1} = s_f)$, edge-level reversibility suggests

$$F(s_i)P_F(s_{i+1}|s_i) = F(s_{i+1})P_B(s_{i+1}|s_i), \quad \text{for all } (s_i, s_{i+1}) \subset t^f.$$

Therefore, for every partial trajectory $t' = (s_k, s_{k+1}, \ldots, s_{k+m}) \subset t^f$, by using the fact that $\forall 0 \leq i \leq k-1, F(s_{i+1}) = \frac{F(s_i)P_F(s_{i+1}|s_i)}{P_B(s_i|s_{i+1})}$, we have

$$F(s_k)\prod_{i=1}^{k} P_F(s_{k+i}|s_{k+i-1}) = F(s_{k+m})\prod_{i=1}^{k} P_B(s_{k+i-1}|s_{k+i}) \tag{19}$$

which suggests the target of SubTB($\lambda$). Hence, it is only required to check whether the edge-level reversibility can be achieved from the Markov chain perspective. In fact, this edge-level reversibility is equivalent to the **detailed balance conditions** (Douc et al., 2018). If detailed balance conditions are satisfied, the probability measures are unique if the corresponding finite-state Markov chain is irreducible and positive recurrent (Douc et al., 2018). Given that flows are also unique in GFlowNets (Bengio et al., 2023), it is required to show whether our MC modeling of GFlowNets achieve unique flows as well. Since flows are unnormalized probability measures, it suffices to show that the Markov transition kernel $P_{0.5}$ yields a irreducible and positive recurrent Markov chain, which is suggested in Prop. **??**. Therefore, DB and SubTB($\lambda$)'s convergence to unique flows is ensured from a Markov chain viewpoint.

For **TB**, we relate to the **Kolmogorov's criterion** (Douc et al., 2018), which is equivalent to the detailed balance conditions for a finite discrete Markov chain like GFlowNets. Kolmogorov's criterion suggest that for any loop $(s_0 = \bar{s}, s_1, \ldots, s_{n-1}, s_n = \bar{s})$, the reversibility at the loop is achieved, i.e.

$$\pi(s_0)\prod_{i=1}^{n} P(s_i|s_{i-1}) = \pi(s_n)\prod_{i=1}^{n} P(s_{i-1}|s_i) \tag{20}$$

if the corresponding finite discrete Markov chain is irreducible and positive recurrent. If one mergers the source state $s_s$ and the final state $s_f$ as Deleu & Bengio (2023), i.e. $s_s = s_f = \bar{s}$, then TB of $P_{0.5}$ targets at

$$F(\bar{s})\prod_{i=1}^{m} P_{0.5}(s_i|s_{i-1}) = F(\bar{s})\prod_{i=1}^{m} P_{0.5}(s_{i-1}|s_i)$$

which translates into

$$\pi(\bar{s})\prod_{i=1}^{m} P_{0.5}(s_i|s_{i-1}) = \pi(\bar{s})\prod_{i=1}^{m} P_{0.5}(s_{i-1}|s_i)$$

for the complete trajectory $t^f = (s_0 = s_s, s_1, \ldots, s_{n-1} = x, s_n = s_f)$ since $\pi(s) = \frac{F(s)}{Z_{\text{state}}}$ is the corresponding probability measure and $P_B(x|s_f) = \frac{R(x)}{\sum_{x' \in \mathcal{X}} R(x')}$. Therefore, the convergence of TB is a necessary condition for Kolmogorov's criterion to hold. Due to the fact that $P_{0.5}$ yields a irreducible and positive recurrent Markov chain (see Prop. **??**), if Kolmogorov's criterion holds, the uniqueness of flows is achieved by the convergence of TB as well. However, since TB is only a necessary condition, this uniqueness is relatively fragile, which potentially contribute to instability of TB training (Madan et al., 2023).

### C.3 Proof for Prop. 4.4

**We first prove the claim that $\alpha$-DB, $\alpha$-SubTB and $\alpha$-TB correspond to edge-level, partial-trajectory-level and trajectory-level reversibility of $P_\alpha$.** The proof is similar to the proof for Thm. 4.2, and we only show the proof for $\alpha$-DB. Recall the reversibility of $\alpha$-DB: for any $(s, s') \in \mathbb{A}$

$$\pi(s)P_\alpha(s'|s) = \pi(s')P_\alpha(s|s'). \tag{21}$$

Since both $P_F$ and $P_B$ are one-directional, Eq. 21 implies

$$\alpha\pi(s)P_F(s'|s) = (1-\alpha)\pi(s')P_B(s|s'). \tag{22}$$

On the other hand, recall $\alpha$-DB, which targets at

$$\alpha F(s)P_F(s'|s) = (1-\alpha)F(s')P_B(s|s'). \tag{23}$$

Plugging $\pi(s) = \frac{F(s)}{Z_{\text{state}}}$ into Equation 23 yields

$$\alpha\pi(s)P_F(s'|s) = (1-\alpha)\pi(s')P_B(s|s')$$

which is identical to Eq. 22. The proofs for $\alpha$-SubTB and $\alpha$-SubTB($\lambda$) are similar.

**Next, we show the convergence of such objectives lead to unique state flows from the Markov chain perspective.** This also follows the proof for Thm. 4.2. The convergence of $\alpha$-DB and $\alpha$-SubTB($\lambda$) to unique flows are derived by the detailed balance conditions, irreducibility and positive recurrence of the Markov chain with $P_\alpha$, whereas that of $\alpha$-TB follows a necessary condition of Kolmogorov's criterion of $P_\alpha$.

**Nevertheless, App. B.3 reveals that the convergence rates to unique flows vary for different $\alpha$ values.** Even though this aspect may not be perfectly reflected by loss curves due to the difference between GFlowNet training and MCMC algorithms, the tuning of $\alpha$ still contributes the training of GFlowNets, as suggested in Prop. 4.5.

### C.4 Proof for Prop. 4.5

Following Def. 4.3, the loss function of $\alpha$-SubTB at the partial trajectory $\mathfrak{t} = (s_k, s_{k+1}, \ldots, s_{k+m})$ is

$$L_{\alpha-\text{SubTB}}(\mathfrak{t}') = \log^2\left(\frac{\alpha^m F(s_k)\prod_{i=1}^m P_F(s_{k+i}|s_{k+i-1})}{(1-\alpha)^m F(s_{k+m})\prod_{i=1}^m P_B(s_{k+i-1}|s_{k+i})}\right).$$

We denote by $P_B(\mathfrak{t}') = \prod_{i=1}^m P_B(s_{k+i-1} \mid s_{k+i1})$. Then, the gradient to $P_F(\mathfrak{t}') = \prod_{i=1}^m P_F(s_{k+i} \mid s_{k+i-1})$ is

$$\frac{\partial L_{\alpha-\text{SubTB}}(\mathfrak{t}')}{\partial P_F(\mathfrak{t}')} = \frac{2}{P_F(\mathfrak{t}')}\log\frac{\alpha^m F(s_k)P_F(\mathfrak{t}')}{(1-\alpha)^m F(s_{k+m})P_B(\mathfrak{t}')}$$

$$= \frac{2}{P_F(\mathfrak{t}')}\log\frac{F(s_k)P_F(\mathfrak{t}')}{F(s_{k+m})P_B(\mathfrak{t}')} + \frac{2m}{P_F(\mathfrak{t}')}\log\frac{\alpha}{1-\alpha}.$$

Meanwhile, taking a gradient of Eq. 9 suggests

$$\frac{\partial L_{\text{SubTB}}(\mathfrak{t}')}{\partial P_F(\mathfrak{t}')} = \frac{2}{P_F(\mathfrak{t}')}\log\frac{F(s_k)P_F(\mathfrak{t}')}{F(s_{k+m})P_B(\mathfrak{t}')}.$$

Therefore, it is direct that

$$\frac{\partial L_{\alpha-\text{SubTB}}(\mathfrak{t}')}{\partial P_F(\mathfrak{t}')} = \frac{\partial L_{\text{SubTB}}(\mathfrak{t}')}{\partial P_F(\mathfrak{t}')} + \frac{2m}{P_F(\mathfrak{t}')}\log\frac{\alpha}{1-\alpha}.$$

# D EXPERIMENTS

In this section, we present the detailed experimental settings, computational resource usage, ablation studies of our $\alpha$-GFNs along with more numerical results and corresponding analysis to support our findings.

## D.1 DETAILED EXPERIMENT SETUPS AND COMPUTATIONAL RESOURCE USAGE

**Set Generation**   We implement $\alpha$-DB, $\alpha$-TB and $\alpha$-FL-DB based on the open-source code of Pan et al. (2023), where setting $\alpha = 0.5$ yields the baselines. Most details of this task follows Pan et al. (2023), including the models and hyperparameters. To be specific, the dimension of the vocabulary (the action space) is 30, 80 and 100 for small, medium and large sets, respectively. The maximum capacity of sets is capped at 20, 60, and 80 for small, medium and large sets. We employ the same intermediate energy function $\mathcal{E}(\cdot)$: energies are sampled uniformly from $[-1, 1]$, and exactly $\frac{|S|}{10}$ elements share identical energy values, resulting in multiple optimal solutions. The GFlowNet agent is parameterized by an MLP with two hidden layers of 256 units and LeakyReLU activations. We generate 16 samples per training step and use the Adam optimizer (Kingma & Ba, 2014) to optimize the objectives. All the objectives are trained with 10,000 steps. The learning rate for $\alpha$-DB and $\alpha$-FL-DB is set to 0.001. For $\alpha$-TB, we apply a learning rate of 0.001 to the MLP parameters and 0.1 to the learnable parameter $Z$, which represents the total amount of flows, for TB-based objectives, which directly follows Pan et al. (2023). The backward policy $P_B$ is set to be uniform. In particular, we notice that the parameter $\epsilon$ in the $\epsilon$-greedy sampling trick varies across different values and set sizes. In some cases, the sampling policy is even totally uniform ($\epsilon = 1$). Therefore, we adopt a unified approach for the setting of $\epsilon$: we first start with $\epsilon = 1$, and linearly anneal $\epsilon$ to 0.05 during the training.

The evaluation metrics are 1) **Modes**: the number of unique samples with reward greater than a fixed threshold. Since Pan et al. (2023) does not mention the reward threshold to define modes, we set the threshold to be 0.25 for small sets, following Jang et al. (2024), and 700,000 for medium and large sets. This metric measures whether the policy effectively explore different high-reward regions. 2) **Top-1000 Reward**: the average reward of unique samples with the top-1000 highest reward, a metric to test the exploitation ability of the policy. 3) **Spearman Corr**: the spearman correlation between the generation probability and the reward in a held-out test set. of size 1000 generated by the initialized policy. We additionally report the **average reward of all samples generated in on-policy evaluation in Fig. 1.**

1. remove the corresponding lines 2.

Following Pan et al. (2023), the evaluation is conducted online and independent of training samples. Training samples are generated with the $\epsilon$-greedy policy, while evaluation samples are generated only with the forward policy $P_F$. Models are trained with 10,000 steps, where the first 9,000 steps are stage 1. Throughout the experiments, we fix the initialization of model weights and use different random seeds for sample generation. All the results are averaged over 10 random seeds, which are integers from 0 to 9. For each of the $\alpha$-GFN objectives, it takes about 10-20 minutes, 25-50 minutes and 30-60 minutes for small, medium and large sets to run an experiment on a cluster consisting of NVIDIA RTX3090, NVIDIA RTX4090 and NVIDIA ADA6000 GPUs.

**Bit Sequence Generation**   Here, we adopt the non-autoregressive version of this task, which is proposed in Tiapkin et al. (2024) and is more difficult than the original version in Malkin et al. (2022). We implement $\alpha$-DB, $\alpha$-TB and $\alpha$-SubTB based on the open-source code of Tiapkin et al. (2024), where setting $\alpha = 0.5$ yields the baselines, and follow the default hyperparameters and setup as Tiapkin et al. (2024). We apply an $\epsilon$-noisy strategy that mixes the GFlowNet forward policy $P_F$ with a uniform distribution $U$, assigning weights of $(1 - \epsilon)$ and $\epsilon$ (with $\epsilon = 0.001$), while using a uniform backward policy $P_B$. The mode set $M$ and test set are constructed as in (Malkin et al., 2022). A Transformer model (Vaswani et al., 2017) with 3 hidden layers, 64-dimensional hidden states, and 8 attention heads (Tiapkin et al., 2024) is employed across

all objectives. Training is performed using the Adam optimizer (Kingma & Ba, 2014) with batch size 16 and learning rate $2 \times 10^{-3}$. For TB-based objectives, the learnable parameter $Z$ uses a learning rate of $10^{-3}$. For SubTB($\lambda$) objectives we set $\lambda = 1.9$. We adopt a gradient clipping norm of 20 to stabilize the training. All models are trained for 50,000 steps, and the reward function is augmented as $\tilde{R}(x) = R(x)^{\beta}$ with $\beta = 2$. To approximate the probability of generating a sample $x \in \mathcal{X}$, which is formulated as $P_{\theta}(x)$, we use the Monte Carlo estimation proposed in (Zhang et al., 2022):

$$P_{\theta}(x) = \mathbb{E}_{P_B(\mathfrak{t}|x)} \frac{P_F(\mathfrak{t})}{P_B(\mathfrak{t}|x)} \approx \frac{1}{N} \sum_{i=1}^{N} \frac{P_F(\mathfrak{t}^i)}{P_B(\mathfrak{t}^i|x)}$$

where $\mathfrak{t}^i \in \mathfrak{T}^{\text{flow}}$ is a trajectory generated by the model. We set $N = 10$, following Tiapkin et al. (2024).

We adopt the evaluation metrics in Tiapkin et al. (2024). To be specific, the evaluation metrics of this task are 1) **Modes**: the number of unique samples with Hamming distance less than 30 to the predefined set $M$, ranging from 0 to 60. 2) **Spearman Corr**: the Spearman correlation between the generation probability of samples and their reward in a predefined test set.

Following the practice in Set Generation, we separate training and evaluation samples. Training samples are generated with the $\epsilon$-greedy policy, while evaluation samples are generated with only the forward policy $P_F$ online. Models are trained with 50,000 steps, where the first 40,000 steps are stage 1. Throughout the experiments, we fix the initialization of model weights and use different random seeds for sample generation. All the results are averaged over 5 random seeds, which are integers from 0 to 4. Each of the setting (k, objective, $\alpha$) takes about 1 day to run on a cluster consisting of NVIDIA RTX3090, NVIDIA RTX4090 and NVIDIA ADA6000 GPUs.

**Molecule Generation** The goal of this task is to generate binders of the sEH (soluble epoxide hydrolase) proteins. In this task, we implement $\alpha$-DB, $\alpha$-FL-DB, $\alpha$-SubTB($\lambda$), $\alpha$-SubTB($\lambda$) and $\alpha$-TB based on the open-source code of Tiapkin et al. (2024) and Pan et al. (2023). We follow the default models and hyperparameters of Tiapkin et al. (2024). We utilize a Message Passing Neural Network (MPNN) with 10 convolution steps and train it via the Adam optimizer (Kingma & Ba, 2014) using a batch size of 4 and a learning rate of $5 \times 10^{-4}$. The reward is exponentiated with $\beta = 4$ to produce $\tilde{R}(x) = R(x)^{\beta}$ as augmentation. An $\epsilon$-greedy mixture of the GFlowNet forward policy $P_F$ and a uniform policy $U$ is applied, i.e. $P = \epsilon U + (1-\epsilon) P_F$ with $\epsilon = 0.05$ is used for sampling. For TB objectives, the learnable parameter $Z$ is initialized at $\log Z = 30$ and is optimized with a learning rate of $5 \times 10^{-4}$. SubTB($\lambda$)-based objectives are trained with $\lambda = 0.99$ as in (Tiapkin et al., 2024). We adopt a gradient clipping norm of 2 for TB to stabilize the training.

The evaluation metrics are 1) **Modes**: the number of unique molecules with reward greater than 7 and Tanimoto similarity less than 0.7; 2) **Top-k Reward**, $k \in \{10, 100, 1000\}$: the average reward of unique samples with the top-k highest reward. 3) **Top-k Similarity**, $k \in \{10, 100, 1000\}$: the average Tanimoto similarity of samples within Top-k Reward. 4) **Spearman Corr**: the Spearman correlation between the probability of sample generation and their reward in a predefined test set.

Following the practice in Set Generation, we separate training and evaluation samples. Training samples are generated with the $\epsilon$-greedy policy, while evaluation samples are generated with only the forward policy $P_F$ online. We follow the defaulted setting in Pan et al. (2023) but adopt a smaller total number of steps 50,000 due to computational budgets, where the first 40,000 steps are stage 1. Training spans 50,000 steps per objective, generating 200,000 molecules for training and 200,000 molecules for evaluation. We repeat each of the settings 4 times to obtain the results. Throughout the experiments, we fix the initialization of model. Since random seed generators for sampling are set to be timestamps in the original codebase, which causes trouble in reproduction of the results, we fix the random seed generators for training with seeds ranging from 0 to 4, which are 5 integers in total, to facilitate reproduction of the results. We follow the implementation in OrderedGNN (Song et al., 2023), making message passing computation reproducible. Each of the objective

settings takes about 1 day to run on a cluster consisting of NVIDIA RTX3090, NVIDIA RTX4090 and NVIDIA ADA6000 GPUs, where we run 5 settings in parallel within a single GPU.

## D.2 ADDITIONAL RESULTS

**Length-controling side-effects.** We show in Fig. 6 how $\alpha$ changes sample length. Because samples in Set Generation and Bit Sequence Generation have fixed lengths, we use Molecule Generation as the representative example.

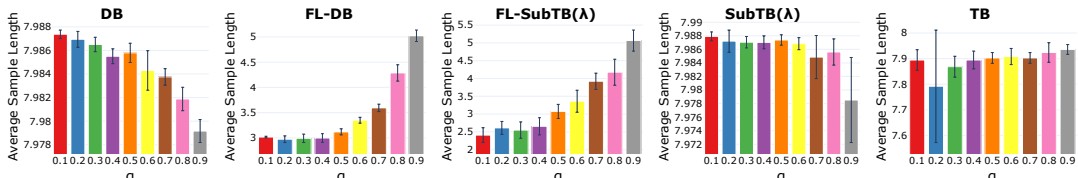

Figure 6: **Average Sample Length** vs $\alpha$ in Molecule Generation across different objectives.

**Dynamics throughout scheduled training.** We visualize how key metrics evolve across training steps under the scheduled scheme (Alg. 1). For Set Generation, see Figs. 7–11; for Bit Sequence Generation, see Figs. 12–14; and for Molecule Generation, see Figs. 15–20.

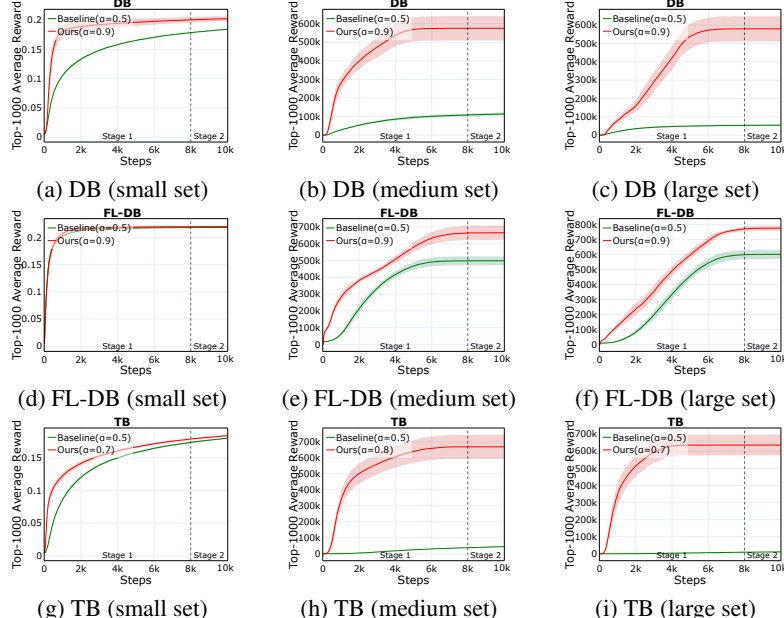

Figure 7: **Top-1000 Average Reward** vs Training Steps in **Set Generation** across different objectives and set sizes.

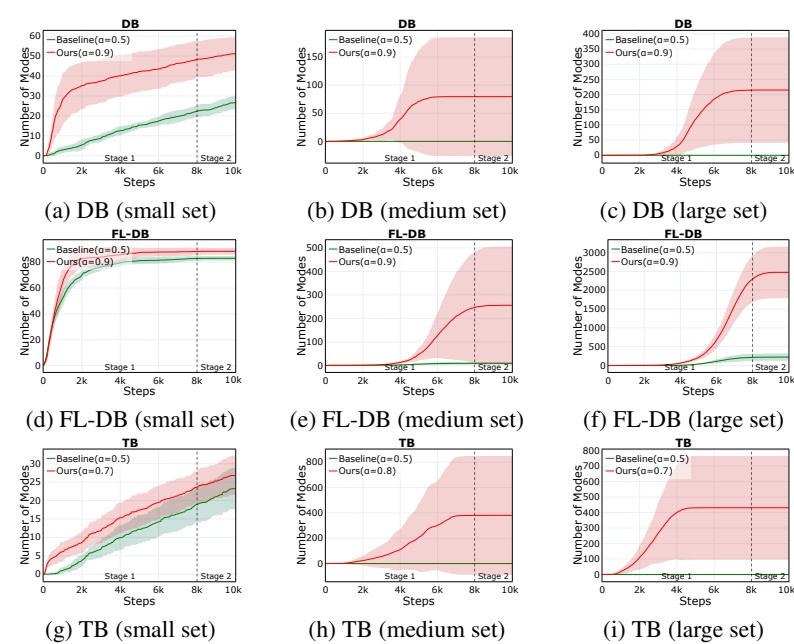

Figure 8: **Number of Modes** vs Training Steps in **Set Generation** across different objectives and set sizes.

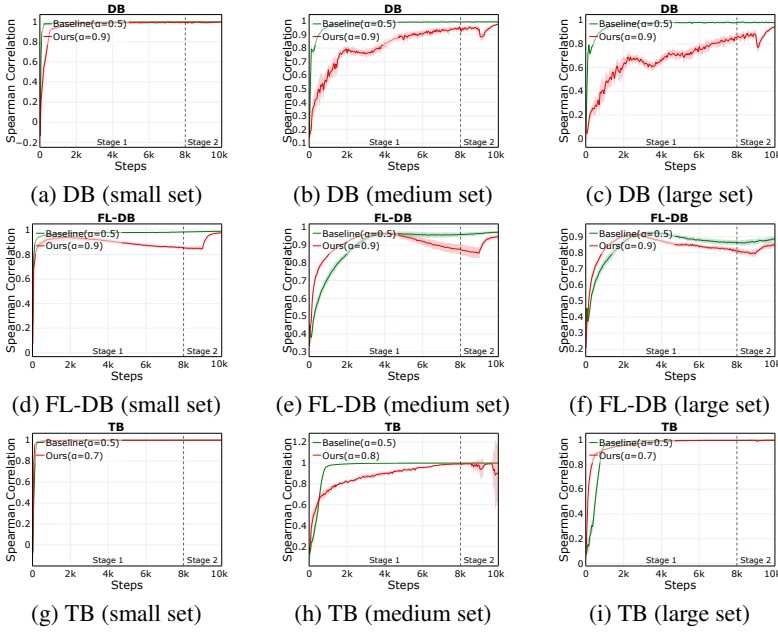

Figure 9: **Spearman Correlation** vs Training Steps in **Set Generation** across different objectives and sizes.

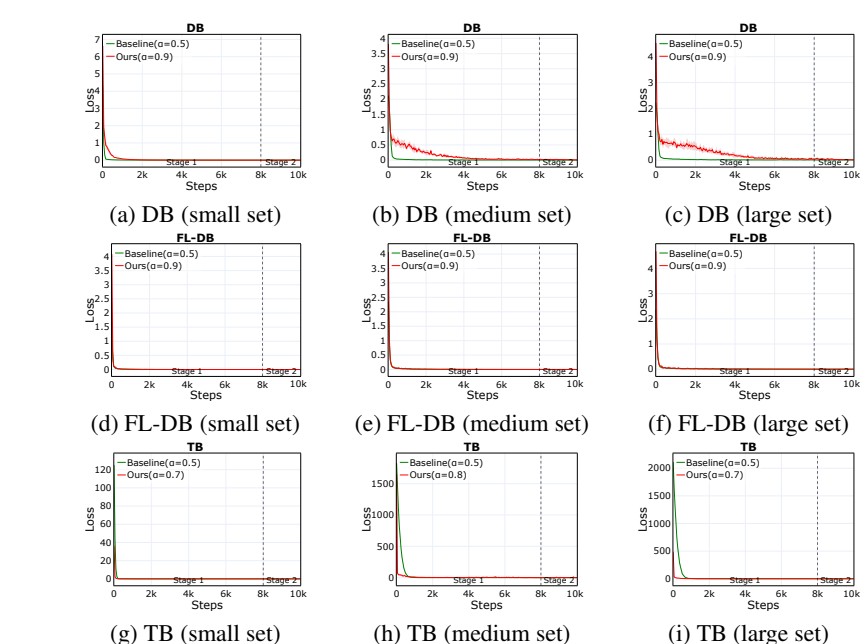

Figure 10: **Loss** vs Training Steps in **Set Generation** across different objectives and set sizes.

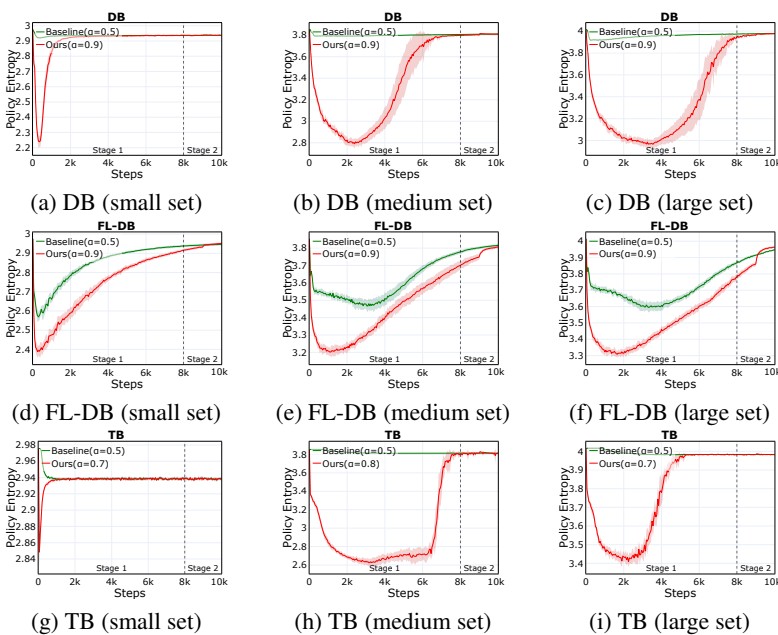

Figure 11: **Forward Policy Entropy** vs Training Steps in **Set Generation** across different objectives and set sizes.

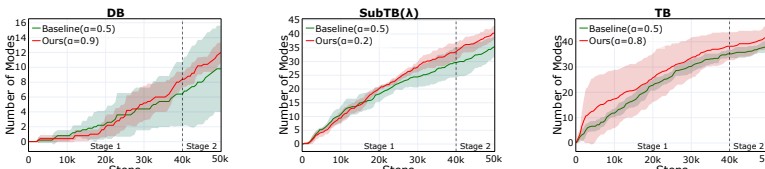

Figure 12: **Number of Modes** vs Training Steps in **Bit Sequence Generation** across different objectives.

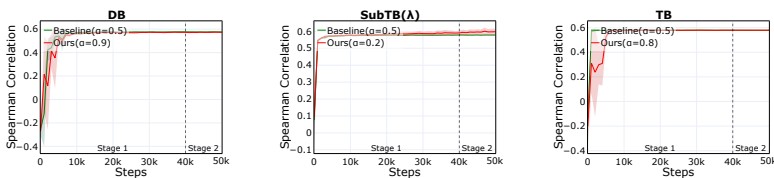

Figure 13: **Spearman Correlation** vs Training Steps in **Bit Sequence Generation** across different objectives.

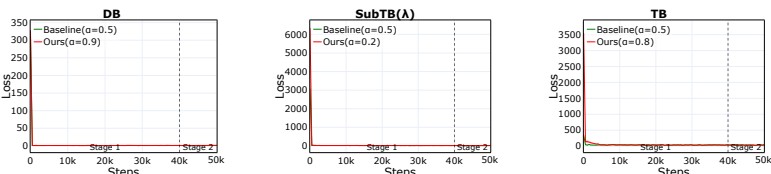

Figure 14: **Loss** vs Training Steps in **Bit Sequence Generation** across different objectives.

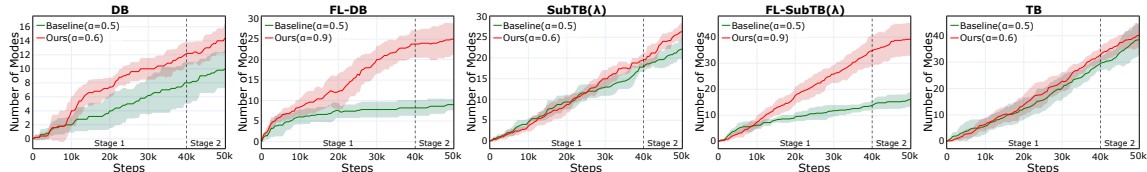

Figure 15: **Number of Modes** vs Training Steps in **Molecule Generation** across different objectives.

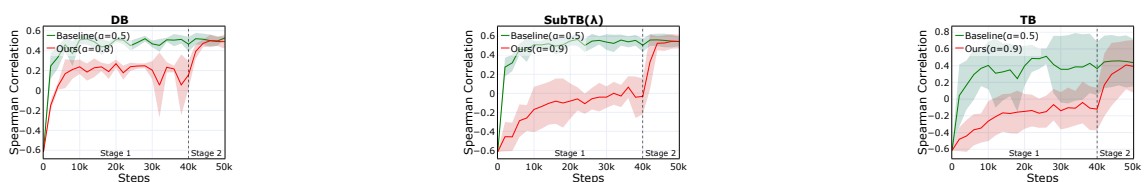

Figure 16: **Spearman Correlation** vs Training Steps in **Molecule Generation** across different objectives. FL-DB and FL-SubTB($\lambda$) are omitted due to their biased target (Silva et al., 2025).

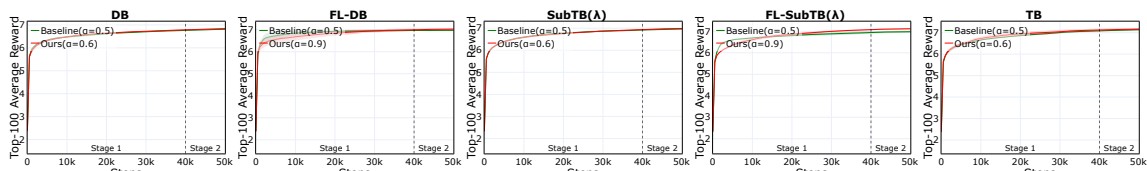

Figure 17: **Top-100 Average Reward** vs Training Steps in **Molecule Generation** across different objectives.

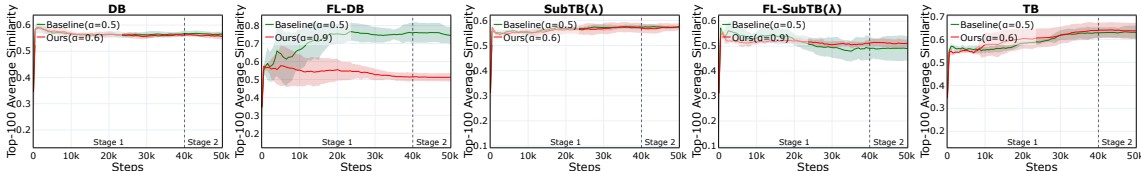

Figure 18: **Top-100 Average Similarity** vs Training Steps in **Molecule Generation** across different objectives.

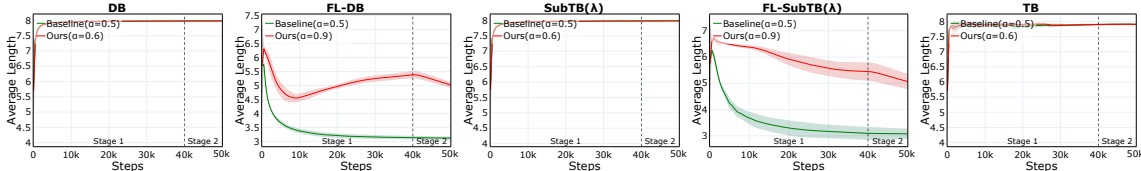

Figure 19: **Average Sample Length** vs Training Steps in **Molecule Generation** across different objectives.

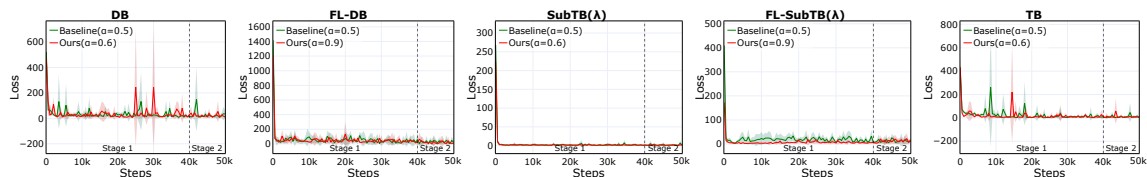

Figure 20: **Loss** vs Training Steps in **Molecule Generation** across different objectives.

## E  ETHICS STATEMENT

This paper presents $\alpha$-GFN, a GFlowNet algorithm that improves the exploration–exploitation trade-off. By advancing GFlowNet theory and methodology, it directly supports scientific applications. We evaluate on widely used public benchmarks to ensure transparency and fairness, and our efficient configurations limit compute and environmental impact, underscoring our dedication to sustainable, reproducible research.

## F  REPRODUCIBILITY STATEMENT

We include implementation and dataset details in Sec. 5.1, full experimental settings in App. D.1, and the complete training procedure in Alg. 1 to ensure reproducibility. We provide all mathematical formulations, implementation details, and configuration settings required for reproducing our results in the paper and appendices.

## G  THE USE OF LARGE LANGUAGE MODELS (LLMs)

LLMs are used solely to assist writing and style polishing. All fundamental research ideas, theoretical derivations, experimental setups, and algorithmic innovations are developed solely by the authors without any LLM assistance. The human researchers exclusively derive the mathematical formulations, establish the proofs, and obtain the experimental results. LLMs did not contribute to the fundamental conceptual development of $\alpha$-GFN or to the core insights about exploration–exploitation trade-offs.

