# OpenReview forum: "$\alpha$-GFN: Generalized Mixing in GFlowNets for Better Exploration-Exploitation Trade-off"
_ICLR.cc/2026/Conference — Submitted to ICLR 2026_

### Official Review · Reviewer_Cy2F · 2025-10-20

**Soundness:** 2
**Presentation:** 2
**Contribution:** 1
**Rating:** 2
**Confidence:** 5

**Summary:**

The paper introduces α-GFlowNets (α-GFNs), a generalization of Generative Flow Networks (GFlowNets) that allows unequal mixing of forward and backward policies using a tunable hyperparameter α ∈ (0, 1). This breaks the traditional assumption of equal weighting and provides explicit control over exploration vs. exploitation during training. The theoretical ground is mostly solid and the method is easy to understand.

**Strengths:**

- Strong Theoretical Innovation:
This paper proposes a generalized framework (α-GFN) that extends traditional “balanced flow” GFlowNets to imbalanced flows via a tunable α parameter. This paper establishes a formal equivalence between standard GFlowNet objectives (DB, SubTB, TB) and the reversibility condition of Markov chains.

- Simple yet Effective Design
The method requires minimal modification—just introducing a scalar α into existing losses (DB, SubTB, TB). The two-stage scheduling algorithm (annealing α from off-center to 0.5) is conceptually simple, easy to implement, and compatible with existing training setups.

- Interesting Secondary Finding
Discovers that α indirectly controls trajectory length, offering new insight for length-controllable GFlowNet sampling, a potential future research direction.

**Weaknesses:**

There are several substantial weaknesses in this paper:

* **W1. Missing baselines.** Although I do not usually emphasize the need for additional baselines, some critical ones are missing here.

  * *Kim et al. (ICLR 2025), "Adaptive Teachers for Amortized Samplers"*
  * *Lau et al. (NeurIPS 2024), "QGFN: Controllable Greediness with Action Values"*

Both methods focus on improving the sampling efficiency and performance of GFlowNets. For instance, the adaptive teacher approach leverages a teacher model to guide exploration more effectively. Since this paper also evaluates mode discovery and average reward as its primary metrics, omitting these stronger baselines (and relying only on TB, DB, and FL-TB) weakens the empirical comparison.

* **W2. Novelty concerns.** The paper presents limited novelty. The main contribution:unequal mixing of forward and backward policies, requires only minimal modifications to existing GFlowNet implementations. Even though a scheduling algorithm is proposed to adjust the value of $\alpha$, this addition does not appear to offer substantial conceptual advancement over prior work.

* **W3. Reproducibility concerns.** The authors do not provide code, and the described method is simple enough to reimplement. However, when I attempted to reproduce the results using the unequal mixing strategy, the performance was significantly worse than the results reported in the paper. While I do not intend to suggest any data fabrication, this discrepancy indicates possible implementation or reporting inconsistencies.

* **W4. Clarity issues.** Several parts of the paper lack clarity. For example, the authors state that “Corresponding to imbalanced flows, unequal weights can enhance the performance of GFlowNets, as illustrated in Fig. 1,” but *Fig. 1* only depicts schematic arrows between forward and backward policies, without any actual illustration of performance enhancement. Moreover, the term **“unique flows”** appears multiple times, yet it is not defined anywhere. Since this terminology is uncommon in the GFlowNet literature, its undefined usage hinders readability and understanding, making the paper hard to follow.

* **W5. Exaggerated results.** The second subfigure of *Fig. 1* claims a **270× improvement** over TB, but neither the task setup nor the precise definition of the “average reward” metric in this context is provided. This lack of detail makes the claimed improvement difficult to verify and potentially misleading.

- **W6. Concerns about performance.** The performance improvements achieved by the proposed method appear to be marginal on both the bit sequence generation and molecule generation tasks. For instance, the results of TB with $\alpha = 0.9$ are only about 1.5× higher than the baseline (TB with $\alpha = 0.5$), whereas methods such as QGFN report nearly 4× improvements over TB under similar settings. Furthermore, the Spearman correlation is lower in several cases, suggesting that the proposed approach may not consistently enhance learning stability.

- **W7. Violation of the flow structure.** Does Equation (3) violate the fundamental principle of GFlowNets? By definition, a GFlowNet performs probabilistic inference—its goal is to learn to sample from a target distribution induced by a given reward function, ensuring that the total inflow to each state equals its total outflow. However, the proposed method introduces an imbalance through the $\alpha$-weighted mixing, which appears to disrupt this flow condition. This raises serious concerns about the theoretical soundness of the approach, as this property is the core foundation of the GFlowNet framework. If the flow structure is indeed preserved, please prove that the global minimization of your objective implies that the learned action policy samples proportionally to $R$.

- **W8. More solid experimental results.** Beyond the number of modes and Spearman correlation, there exist additional quantitative metrics to assess the performance of GFlowNets, such as ELBO and EUBO, as reported in “Adaptive Teachers for Amortized Samplers.” Including these metrics would make the evaluation more thorough and convincing. Moreover, the current experimental setups appear relatively simple and may not fully demonstrate the proposed method’s effectiveness. The paper would be significantly strengthened by scaling the experiments to more realistic and complex scenarios, such as language model fine-tuning tasks (“Amortizing Intractable Inference in Large Language Models”) or listwise recommendation tasks (“Generative Flow Network for Listwise Recommendation”).


- **W9. Inconsistent experimental setup.** In set generation task, three baselines are reported: DB, FL-DB and TB. In bit sequence generation tasl, three different baselines are reported: DB, SubTB and TB. As for molecule generation task, five baselines are reported: DB, FL-DB, SubTB, FL-SubTB and TB. Such inconsistencies make it difficult to perform a fair comparison and to clearly assess the effectiveness of the proposed method. The authors are encouraged to align the experimental setups across tasks to ensure consistency and fairness in evaluation.

Minor mistakes:
-  **Mistakes in formulations.** The equation on *line 964* contains an error in the term $$\frac{1}{2}P_B(s_{k+i-1}\mid s_{k+i}).$$ Additionally, there is a typographical error in *Formulation (21)* on *line 1039*, where $P_{(\alpha)}$  should be corrected to $P_{\alpha}$.

- **Mismatch with previous works.** The number of modes reported for the baseline methods in the molecule generation task is notably lower than that reported in previous studies. While I understand that differences in experimental setups can lead to variations in results, could you further clarify why your reported results are significantly worse?

**Questions:**

I think I've said all that needs to be said elsewhere.

---

> ### Author Response · Authors · 2025-11-28
> **Response to Reviewer Cy2F (1/7)**
>
> We thank the reviewer for recognizing the theoretical novelty of α‑GFN, its simple yet effective design, and the secondary finding that it enables trajectory‑length control. Below, we address each of the reviewer’s concerns point by point.
>
> >**[W1]** Missing baselines. Add baselines like Adaptive Teachers and QGFN.
>
> We appreciate the reviewer’s suggestion to include Adaptive Teacher and QGFN as baselines. Our choice of main baselines in the paper was primarily driven by the specific phenomenon we aim to study.
>
> - First, **our paper focuses on how introducing $\alpha$ changes flow balancing and the corresponding exploration-exploitation trade-offs in GFlowNets**. For this reason, we deliberately chose DB, TB, SubTB and their forward-looking variants as our main baselines, since they are the canonical objectives that explicitly optimize for balanced flows. In contrast, Adaptive Teacher and QGFN mainly modify the training dynamics and sampling efficiency (e.g., via teacher policies or Q‑guided mixing) on top of a given GFlowNet objective. They are therefore complementary to, rather than direct alternatives to, the way $\alpha$-GFN introduces flow imbalance. To keep the scope focused, the main paper compares $\alpha$-GFN against these standard balanced-flow objectives.
>
> - Second, **to directly address the reviewer’s concern, we additionally evaluate $\alpha$-GFN on top of Adaptive Teacher and QGFN on the sEH benchmark, using their official codebases**. (Note that the sEH tasks differ in these two codebases, and that their settings also differs from ours). For Adaptive Teacher, we apply $\alpha$-GFN only to the student objective while keeping the teacher flow unbiased with $\alpha$=0.5, and leave all other settings as the default settings of the codebase. For QGFN, we adopt the $p$-greedy TB configuration and replace the underlying TB loss with its $\alpha$-GFN counterpart, leaving all other hyperparameters unchanged except for the reward/similarity thresholds to align with the paper. The resulting performance of Adaptive Teacher+$\alpha$-GFN and QGFN+$\alpha$-GFN is summarized in the following, which shows that our method is compatible with and beneficial to these stronger baselines. Corresponding evaluation metrics are adopted from the Adaptive Teacher paper and the QGFN paper respectively.
>
> **Table 1: $\alpha$-GFN performance under adaptive teacher.**
> | Metric / $\alpha$ values | Ours                | Baseline  |
> |--------------------------|--------------------|-----------------|
> | Modes $\uparrow$         | **110.00±2.55**    | 103.50±5.07     |
> | ELBO                     | 5.72±0.17          | 5.75±0.09       |
> | EUBO                     | 9.12±0.89          | 6.75±0.94       |
> | Pearson Corr             | 0.84±0.06          | 0.82±0.04       |
>
> It is observed that $\alpha$-GFN finds more modes than Adaptive Teacher.
>
> **Table 2: $\alpha$ vs. performance under QGFN.**
> | Metric / $\alpha$ values | Ours              | Baseline              |
> |--------------------------|------------------|------------------|
> | Modes$\uparrow$               | **613.40±269.85**    | 435.80±176.90    |
> | Average Reward$\uparrow$               | 0.81±0.06        | 0.66±0.22        |
>
> $\alpha$-GFN also surpasses the QGFN baseline. **In short, $\alpha$-GFN remains effective in these two baselines**.

---

> ### Author Response · Authors · 2025-11-28
> **Response to Reviewer Cy2F (2/7)**
>
> > **[W2]** Novelty concerns: mixing policies by $\alpha$ is too simple even though there's a scheduling algorithm, offering limited conceptual advancement.
>
> We respectfully disagree with the assessment that the novelty of our work is limited, and summarize the main contributions of our work as follows:
> 1. **We present a comprehensive theoretical study of the two frameworks, GFlowNets and Markov chains**. This study deepens the connection between the two frameworks (see Appendix B). Combined with recent works linking GFlowNets to reinforcement learning (RL) [1,2,3], this work also shows a possible connection between traditional Markov chain theories and maximum entropy reinforcement learning, opening corresponding future research directions.
> 2. **We show a benefit of such theoretical connections by introducing $\alpha$-GFN, bridging the gap between the practice of GFlowNets and Markov chains**. We show that certain Markov chain properties can be reflected on the training of $\alpha$-GFNs through a gradient perspective in Section 4.4 (the exploration-exploitation trade-offs), and that such properties can lead to performance gains in experiments across various domains. In other words, the design of $\alpha$-GFN algorithms may be relatively simple, but the theories behind such a design, especially the convergence and validity of $\alpha$-GFNs and the link between GFlowNets and Markov chains, are not.
> 3. **From the GFlowNet perspective, we reveal the phenomenon that imbalanced flows matter**. As discussed in Related Works, previous works in the GFlowNet literature focus on the balance of flows. However, we show that in terms of mode discovery, it is not necessary to have balanced flows. Properly imbalanced flows can boost the performance of GFlowNets (see Table 1, Table 2 and Table 3), and we show that the reason behind such a phenomenon is deeply rooted in Markov chain theories.
> 4. **We not only show the empirical performance gains with $\alpha$-GFNs, but also show the sensitivity and side-effects of $\alpha$**. Apart from the performance gains, we detail in Figure 5 the length-controlling side-effect of our approach, and in Table 4 we show that $\alpha$ is not very senitive in certain scenarios. These aspects, along with $\alpha$-GFN's performance in Tables 1-3, imply the effectiveness and robustness of our approach.
>
> References
>
> [1] Tiapkin, Daniil, et al. "Generative flow networks as entropy-regularized rl." International Conference on Artificial Intelligence and Statistics. PMLR, 2024.
>
> [2] Mohammadpour, Sobhan, et al. "Maximum entropy gflownets with soft q-learning." International Conference on Artificial Intelligence and Statistics. PMLR, 2024.
>
> [3] Deleu, Tristan, et al. “Discrete Probabilistic Inference as Control in Multi-path Environments.” The 40th Conference on Uncertainty in Artificial Intelligence, 2024.
>
> > **[W3]**  Reproducibility concerns. No submitted codes?
>
> Regarding reproducibility, we would like to clarify that the **complete implementation of our method** is already provided in the anonymous **supplementary material** accompanying the submission, and can be used to exactly reproduce the reported results.
>
> > **[W4]** Clarity issues. No performance showcase in Fig.1? What are unique flows?
>
> - In Fig.1 (Right), we present the average reward of all samples generated in the large set setting of Set Generation. Fig.1 (Left) is an illustration of how $\alpha$-GFN is designed: placing $\alpha$ in the forward policy, and biasing the backward policy with weight $(1-\alpha)$. **It is worth noting that Fig.1 presents both the visualization of $\alpha$-GFN and the performance gains in terms of average reward of all samples**.
>
> - The term "unique flows" stems from GFlowNet Foundations [1], where the term "unique Markovian flow" is used (though in the case of $\alpha$-GFNs, the flows may not satisfy the definition of Markovian flows in [1]). **We clarify the uniqueness of flows in Proposition 4.4 of the paper to align with the uniqueness of Markovian flow functions in Proposition 18, Proposition 19, Proposition 21 and Proposition 23 of [1]** to ground the paper theoretically. We discuss in Appendix C.3 that such uniqueness is a natural result of the corresponding Markov chain properties.
>
> Reference
>
> [1] Bengio, Yoshua, et al. "Gflownet foundations." Journal of Machine Learning Research 24.210 (2023): 1-55.
>
> > **[W5]** Exaggerated results. Neither the task setup nor the precise definition of the "average reward" is proivded.
>
> - First of all, in the caption of Fig. 1 we already specify the setup as the large sets configuration of the Set Generation task. For further details on the exact set sizes and corresponding settings, please refer to Appendix D.
> - "average reward" is defined as the average reward of all samples generated during the on-policy evaluations. We've revised the paper accordingly, adding descriptions of this metric. Thank you for pointing this out!

---

> ### Author Response · Authors · 2025-11-28
> **Response to Reviewer Cy2F (3/7)**
>
> > **[W6]** $\alpha$-GFN does not show better performance than QGFN. A drop in Spearman correlations also challenges the learning stability of $\alpha$-GFNs.
>
> We respectfully disagree with this.
> - First, we emphasize that **$\alpha$-GFN and QGFN address orthogonal aspects of GFlowNet training**. QGFN modifies the sampling policy via Q‑guided greediness, whereas $\alpha$-GFN changes the underlying flow-balance condition through an $\alpha$-controlled imbalance. Directly comparing our gains to the $4\times$ improvements reported by QGFN is therefore not an apples-to-apples comparison. To further clarify this point, in the additional experiments reported in our response to **[W1]** (see Table 2 in the response), we apply $\alpha$-GFN on top of QGFN and observe consistent additional performance gains. This shows that $\alpha$-GFN is complementary to methods like QGFN rather than a competing replacement, and can in fact enhance their performance when combined.
> - Next, the design of $\alpha$-GFN violates the flow structure when $\alpha \neq 0.5$. In such a case, the trained forward policy may mismatch with the reward function naturally, leading to a drop in correlation metrics as detailed in Fig. 2. **However, we find empirically that $\alpha$-GFNs find more modes than vanilla GFlowNets even though their correlation metrics are relatively lower sometimes**. Take Fig. 3 as an example, which visualizes the Spearman correlation dynamics in large sets. In terms of modes, the best $\alpha$ here for DB, FL-DB and TB are 0.9, 0.9 and 0.7 respectively, all of which have lower Spearman correlations than vanilla GFlowNets. However, their number of modes is significantly greater than vanilla GFlowNets'. As shown in Table 1, $\alpha$-GFN finds at least 1000% more modes than vanilla GFlowNets in large sets. In Bit Sequence Generation and Molecule Generation, the cases are similar: $\alpha$-GFN sometimes shows lower Spearman correlations, but a higher number of modes. **These results also suggest that Spearman correlations and the number of modes seem not tightly coupled**, indicating that correlation alone may not be a reliable proxy for mode discovery. **By noticing that the goal of GFlowNets in discovery tasks is to find more distinct high-reward modes, we argue that $\alpha$-GFNs show better performance than vanilla GFlowNets** even though they do not fit the reward distribution as well as vanilla GFlowNets.
>
> > **[W7]** $\alpha$-GFNs violate the flow structure and does not satisfy the target of vanilla GFlowNets, raising serious concerns of theoretical soundness.
>
> From a theoretical viewpoint, **$\alpha$-GFN introduces systematic flow imbalance and thus departs from the standard flow-matching structure of GFlowNets**. Consequently, when $\alpha \neq 0.5$, the common target of vanilla GFlowNets, sampling exactly from the distribution induced by a given reward function, may not be satisfied. Nevertheless, the convergence of $\alpha$-GFN can still be established via its link to Markov chain theories. This convergence under imbalanced flows also reveals a deeper connection between GFlowNets and Markov chains, as detailed in Appendix B. Moreover, we show that this controlled imbalance yields a practical benefit: flexible exploration–exploitation trade-offs (Section 4.4, Tables 1–3 and Figure 3), and can enhance the performance of GFlowNets. In short, although $\alpha$-GFNs with $\alpha \neq 0.5$ do not satisfy the common target of GFlowNets, they can still converge and improve performance, and the underlying theory further deepens the connection between GFlowNets and Markov chains.

---

> ### Author Response · Authors · 2025-11-28
> **Response to Reviewer Cy2F (4/7)**
>
> > **[W8]** More evaluation metrics like ELBO/EUBO in Adaptive Teachers? Can $\alpha$-GFNs scale up?
>
> Thank you for your comments on evaluation metrics and scale-up experiments!
> - We add EUBO/ELBO metrics, retrain $\alpha$-GFN in Set Generation and Molecule Generation with 5 seeds (Bit Sequence Generation is omitted due to computational budgets), and report the complete results in the following tables. Since there is no oracle distribution in Set Generation, we use the generated held-out test set to compute EUBO (this practice may introduce inaccuracy sometimes, e.g. EUBO values may be smaller than ELBO). We summarize the complete results as follows:
>
> **Table 1, Set Generation re-evaluated with ELBO/EUBO**
>
> | Set Size | Objective | TB |  | FL-DB |  | DB |  |
> |---------|---------------------|------|------|---------|------|--------|------|
> |         | **Metric** | **Baseline** | **Ours** | **Baseline** | **Ours** | **Baseline** | **Ours** |
> | **Small**  | Modes$\uparrow$  | 23.20 $\pm$ 2.39 | **24.80 $\pm$ 3.27**  | 83.00 $\pm$ 2.45 | **88.60 $\pm$ 2.19**  | 24.80 $\pm$ 2.59 | **55.80 $\pm$ 13.92**  |
> |  | Top-1000 Reward$\uparrow$  | **0.18 $\pm$ 0.00** | **0.18 $\pm$ 0.00**  | **0.22 $\pm$ 0.00** | **0.22 $\pm$ 0.00**  | 0.18 $\pm$ 0.00 | **0.20 $\pm$ 0.01**  |
> |  | Top-1000 Similarity$\downarrow$  | **0.69 $\pm$ 0.00** | **0.69 $\pm$ 0.00**  | **0.73 $\pm$ 0.00** | 0.74 $\pm$ 0.00  | **0.69 $\pm$ 0.00** | 0.72 $\pm$ 0.01  |
> |  | Spearman Corr  | 1.00 $\pm$ 0.00 | 1.00 $\pm$ 0.00  | 0.99 $\pm$ 0.00 | 0.97 $\pm$ 0.01  | 1.00 $\pm$ 0.00 | 1.00 $\pm$ 0.00  |
> |  | ELBO  | -0.01 $\pm$ 0.01 | 0.29 $\pm$ 0.02  | -6.93 $\pm$ 0.18 | -3.78 $\pm$ 0.80  | 0.01 $\pm$ 0.05 | 0.57 $\pm$ 0.05  |
> |  | EUBO  | 0.01 $\pm$ 0.03 | 0.31 $\pm$ 0.03  | -7.24 $\pm$ 0.13 | -4.10 $\pm$ 0.86  | -0.00 $\pm$ 0.04 | 0.61 $\pm$ 0.03  |
> | **Medium**  | Modes$\uparrow$  | 0.00 $\pm$ 0.00 | **499.00 $\pm$ 427.07**  | 14.20 $\pm$ 10.06 | **118.60 $\pm$ 45.88**  | 0.00 $\pm$ 0.00 | **31.40 $\pm$ 34.06**  |
> |  | Top-1000 Reward$\uparrow$  | 44854.41 $\pm$ 1480.97 | **703904.36 $\pm$ 39194.70**  | 507462.73 $\pm$ 47471.45 | **638688.10 $\pm$ 24711.41**  | 110427.38 $\pm$ 5337.25 | **552020.09 $\pm$ 59210.23**  |
> |  | Top-1000 Similarity$\downarrow$  | **0.69 $\pm$ 0.00** | 0.86 $\pm$ 0.01  | **0.78 $\pm$ 0.01** | 0.81 $\pm$ 0.01  | **0.71 $\pm$ 0.00** | 0.80 $\pm$ 0.01  |
> |  | Spearman Corr  | 1.00 $\pm$ 0.00 | 0.86 $\pm$ 0.30  | 0.97 $\pm$ 0.01 | 0.93 $\pm$ 0.01  | 0.99 $\pm$ 0.00 | 0.97 $\pm$ 0.01  |
> |  | ELBO  | 0.03 $\pm$ 0.04 | 1.46 $\pm$ 0.23  | -26.98 $\pm$ 6.03 | -5.67 $\pm$ 2.54  | -0.20 $\pm$ 0.19 | 2.00 $\pm$ 0.15  |
> |  | EUBO  | 0.03 $\pm$ 0.15 | 4.08 $\pm$ 6.11  | -26.98 $\pm$ 6.40 | -5.77 $\pm$ 2.41  | -0.25 $\pm$ 0.14 | 2.25 $\pm$ 0.20  |
> | **Large**  | Modes$\uparrow$  | 0.00 $\pm$ 0.00 | 591.60 $\pm$ 210.56  | 247.60 $\pm$ 147.17 | 2239.20 $\pm$ 169.80  | 0.00 $\pm$ 0.00 | 355.80 $\pm$ 319.84  |
> |  | Top-1000 Reward$\uparrow$  | 11754.28 $\pm$ 230.91 | **683702.46 $\pm$ 32706.82**  | 593090.24 $\pm$ 67865.24 | **768544.60 $\pm$ 6904.37**  | 58086.54 $\pm$ 5216.95 | **624721.77 $\pm$ 55336.88**  |
> |  | Top-1000 Similarity$\downarrow$  | **0.74 $\pm$ 0.00** | 0.87 $\pm$ 0.01  | **0.85 $\pm$ 0.01** | 0.87 $\pm$ 0.00  | **0.76 $\pm$ 0.00** | 0.88 $\pm$ 0.01  |
> |  | Spearman Corr  | 1.00 $\pm$ 0.00 | 1.00 $\pm$ 0.00  | 0.90 $\pm$ 0.02 | 0.85 $\pm$ 0.02  | 0.98 $\pm$ 0.00 | 0.94 $\pm$ 0.00  |
> |  | ELBO  | 0.00 $\pm$ 0.02 | 1.15 $\pm$ 0.07  | -24.20 $\pm$ 1.73 | -30.24 $\pm$ 5.32  | -0.34 $\pm$ 0.18 | 2.65 $\pm$ 0.25  |
> |  | EUBO  | -0.01 $\pm$ 0.14 | 1.26 $\pm$ 0.13  | -22.36 $\pm$ 1.48 | -28.56 $\pm$ 5.05  | -0.35 $\pm$ 0.09 | 2.91 $\pm$ 0.23  |

---

> ### Author Response · Authors · 2025-11-28
> **Response to Reviewer Cy2F (5/7)**
>
> **Table 2: Molecule Generation re-evaluated with ELBO/EUBO**
> |                               | DB             |                | FL-DB          |                | SubTB($\lambda$) |                | FL-SubTB($\lambda$) |                | TB              |                |
> | ----------------------------- | -------------- | -------------- | -------------- | -------------- | -------------- | -------------- | ----------------- | -------------- | --------------- | -------------- |
> |                               | Baseline       | Ours           | Baseline       | Ours           | Baseline       | Ours           | Baseline          | Ours           | Baseline        | Ours           |
> | Modes$\uparrow$                 | 10.40 $\pm$ 2.70 | **11.40 $\pm$ 4.28** | 8.60 $\pm$ 2.70  | **25.00 $\pm$ 4.58** | 22.40 $\pm$ 3.51 | **27.20 $\pm$ 3.11** | 16.60 $\pm$ 3.65    | **36.60 $\pm$ 7.47** | 37.20 $\pm$ 3.11  | **37.40 $\pm$ 3.21** |
> | Top-1000 Reward$\uparrow$       | 6.32 $\pm$ 0.02  | 6.32 $\pm$ 0.02  | **6.71 $\pm$ 0.03**  | 6.51 $\pm$ 0.12  | 6.52 $\pm$ 0.02  | 6.52 $\pm$ 0.02  | 6.40 $\pm$ 0.04     | **6.50 $\pm$ 0.06**  | 6.63 $\pm$ 0.01   | **6.66 $\pm$ 0.03**  |
> | Top-1000 Similarity$\downarrow$ | **0.55 $\pm$ 0.00**  | 0.56 $\pm$ 0.00  | **0.63 $\pm$ 0.02**  | 0.53 $\pm$ 0.06  | 0.56 $\pm$ 0.00  | 0.56 $\pm$ 0.00  | 0.50 $\pm$ 0.03     | **0.50 $\pm$ 0.01**  | **0.58 $\pm$ 0.00**   | 0.59 $\pm$ 0.02  |
> | Spearman Corr                 | 0.48 $\pm$ 0.08  | 0.47 $\pm$ 0.07  | - | - | 0.59 $\pm$ 0.02  | 0.54 $\pm$ 0.17  | - | - | 0.06 $\pm$ 0.61   | 0.44 $\pm$ 0.14  |
> | ELBO| 0.24 $\pm$ 1.31  | -0.13 $\pm$ 1.72 | 0.62 $\pm$ 1.64  | -0.48 $\pm$ 2.96 | -0.08 $\pm$ 0.99 | 0.23 $\pm$ 1.17  | 1.25 $\pm$ 1.91     | -0.77 $\pm$ 5.68 | -7.03 $\pm$ 15.76 | 0.36 $\pm$ 1.35  |
> | EUBO| -6.24 $\pm$ 2.25 | -5.51 $\pm$ 3.03 | 32.59 $\pm$ 2.66 | 27.74 $\pm$ 3.32 | -8.87 $\pm$ 2.67 | -9.79 $\pm$ 0.76 | 21.24 $\pm$ 2.46    | 20.01 $\pm$ 1.53 | 14.61 $\pm$ 24.74 | -4.66 $\pm$ 3.20 |
>
> - Meanwhile, we carry out a **scale-up** test for $\alpha$-GFNs by applying them to the LLM reasoning scenario of FlowRL [1]. To be specific, we apply $\alpha$-GFN with $\alpha \in \{0.1,0.5,0.9\}$ to the objective of FlowRL without any scheduling (fixing $\alpha$=$\alpha_0$ throughout the whole training process).Then, we use the same training/evaluation strategy as FlowRL on the math domain with Qwen2.5-3B-Instruct and the official implementation of FlowRL in the VeRL [2] recipe for 200 steps (the maximum response length is set to be 8k). We report the performance on mathematical reasoning tasks as per FlowRL in the following:
>
> **Table 3: $\alpha$-GFN's performance on FlowRL**
>
> | Benchmark                    | $\alpha$ = 0.1 | $\alpha$ = 0.5 | $\alpha$ = 0.9 |
> |-|-|-|-|
> | AIME2024       |   0.054167  |   **0.056250**  |   **0.056250**  |
> | AIME2025       |   0.033333  |   0.031250  |   **0.045833**  |
> | AMC23          |   0.385937  |   **0.542188**  |   0.496875  |
> | Math500        |   **0.618875**  |   0.587375  |   0.547500  |
> | Minerva        |   **0.240119**  |   0.232537  |   0.186351  |
> | Olympiabench   |   **0.273090**  |   0.243694  |   0.231825  |
>
> A clear trend is observed in Math500, Minerva and Olympiabench: as $\alpha$ decreases, the model benefits from better exploration. In challenging tasks like AIME2024 and AIME2025, enhanced exploitation with larger $\alpha$ values helps. AMC23 lies between these two regimes: the best performance is achieved at the intermediate value $\alpha=0.5$. **These results show that the exploration-exploitation trade-offs with $\alpha$ scale up to various domains**.
>
> To better understand how $\alpha$ works in this case, we investigate the training metrics at step 200 with a focus on the two: (1)average reward and (2) the KL divergence between the trained policy and the reference policy (termed `ref_kl`), i.e. the average of $\pi_{ref}-\pi_{\theta}$, and summarize them in what follows:
>
> **Table 4: how $\alpha$-GFN changes the dynamics of FlowRL**
> |Metric|$\alpha$ = 0.1|$\alpha$ = 0.5|$\alpha$ = 0.9|
> |-|-|-|-|
> |Average Reward|**-0.506**|-0.399|-0.431|
> |`ref_kl`|-0.298|-0.2111|**-0.680**|
>
> It can be seen above that $\alpha=0.1$ recieves relatively low reward due to the exploration, while the policy trained with $\alpha=0.9$ is significantly larger than its reference policy and also achieves higher reward than $\alpha=0.1$ due to the exploitation. These results further validate the analysis on exploration-exploitation trade-offs in Section 4.4.
>
> References
> [1] Zhu, Xuekai, et al. "Flowrl: Matching reward distributions for llm reasoning." arXiv preprint arXiv:2509.15207 (2025).

---

> ### Author Response · Authors · 2025-11-28
> **Response to Reviewer Cy2F (6/7)**
>
> > **[W9]** Inconsistent experimental setup. Add SubTB and FL-SubTB in Set Generation, and add FL-DB and FL-SubTB in Bit Sequence Generation for better comparison
>
> We appreciate the reviewer’s concern about the inconsistency across tasks. Our choices were mainly driven by computational efficiency and by prior empirical evidence.
>
> - **In preliminary experiments of Set Generation, we found that SubTB is significantly less efficient than DB/FL-DB/TB on the large set benchmark, and [1] reports that FL-DB is the strongest-performing objective across various set sizes in this task**. Hence, in the main paper we focused on DB, FL-DB, and TB, which offer a balance between performance comparison and computational cost. To address the reviewer’s concern and complete the picture, we now additionally report SubTB and FL-SubTB results on the small and medium sets (experiments are run with 5 seeds due to computational budget constrains). Experiments on the large set are omitted purely due to computational cost.
>
> **Table 1, Set Generation, SubTB($\lambda$) and FL-SubTB($\lambda$)**
>
> | Set Size | Objective | SubTB |  | FL-SubTB |  |
> |---------|---------------------|------|------|---------|------|
> |         | Metric | Baseline | Ours | Baseline | Ours |
> | **Small**  | Modes$\uparrow$  | 90.00 $\pm$ 0.00 | 90.00 $\pm$ 0.00  | 89.60 $\pm$ 0.89 | **90.00 $\pm$ 0.00**  |
> |  | Top-1000 Reward$\uparrow$  | 0.22 $\pm$ 0.00 | 0.22 $\pm$ 0.00  | 0.22 $\pm$ 0.00 | 0.22 $\pm$ 0.00  |
> |  | Top-1000 Similarity$\downarrow$  | 0.74 $\pm$ 0.00 | 0.74 $\pm$ 0.00  | 0.74 $\pm$ 0.00 | 0.74 $\pm$ 0.00  |
> |  | Spearman Corr  | 1.00 $\pm$ 0.00 | 1.00 $\pm$ 0.00  | 1.00 $\pm$ 0.00 | 0.99 $\pm$ 0.00  |
> |  | ELBO  | 3.31 $\pm$ 0.20 | 3.61 $\pm$ 0.26  | -4.57 $\pm$ 0.50 | -5.09 $\pm$ 0.09  |
> |  | EUBO  | 6.23 $\pm$ 0.09 | 6.44 $\pm$ 0.03  | -3.37 $\pm$ 0.33 | -4.12 $\pm$ 0.14  |
> | **Medium**  | Modes$\uparrow$  | 66.00 $\pm$ 10.61 | **443.00 $\pm$ 249.81**  | 16.00 $\pm$ 8.60 | **258.60 $\pm$ 296.32**  |
> |  | Top-1000 Reward$\uparrow$  | 607960.89 $\pm$ 9605.13 | **706573.85 $\pm$ 25780.99**  | 531806.17 $\pm$ 14738.45 | **655796.05 $\pm$ 68334.69**  |
> |  | Top-1000 Similarity$\downarrow$  | **0.80 $\pm$ 0.00** | 0.84 $\pm$ 0.01  | **0.78 $\pm$ 0.00** | 0.87 $\pm$ 0.01  |
> |  | Spearman Corr  | 0.99 $\pm$ 0.00 | 0.98 $\pm$ 0.00  | 0.97 $\pm$ 0.01 | 0.78 $\pm$ 0.03  |
> |  | ELBO  | -8.72 $\pm$ 0.52 | -6.50 $\pm$ 0.45  | -14.25 $\pm$ 1.50 | -3.02 $\pm$ 1.06  |
> |  | EUBO  | -1.66 $\pm$ 0.47 | -1.20 $\pm$ 0.16  | -7.05 $\pm$ 1.55 | 11.51 $\pm$ 3.07  |

---

> ### Author Response · Authors · 2025-11-28
> **Response to Reviewer Cy2F (7/7)**
>
> - Previous works on forward-looking GFlowNets [1,2] did not report FL variants on Bit Sequence Generation. Since forward-looking objectives require intermediate rewards (which, to our knowledge, was previously not defined in this task), we now implement these objectives  using the following intermediate reward:
>  $$r_t^{\mathrm{inter}} = -\beta \bigl(d_t(x) - d_{t-1}(x)\bigr)$$
>  where
>  $d_t(x) = \min_{m \in M} \sum_{j \in I_t} \mathbf{1}\bigl[x(j) \ne m(j)\bigr]$ is the partial Hamming distance between the current sample and modes.
> We run experiments on the more computationally affordable settings (k=4,6,8,10). The corresponding FL-DB and FL-SubTB results are included below to provide a consistent comparison across objectives.
>
> **Table 2, Bit Sequence, FL-DB and FL-SubTB($\lambda$)**
>
> ||Objective| FL-DB| | FL-SubTB($\lambda$)| |
> |-|-|-|-|-|-|
> | **k** | Metric | Baseline | Ours | Baseline | Ours |
> | **4** | Modes$\uparrow$ | 57.60 $\pm$ 1.34 | **59.20 $\pm$ 0.45** | 58.80 $\pm$ 1.10 | **59.40 $\pm$ 0.55** |
> |  | Spearman | 0.57 $\pm$ 0.00 | 0.57 $\pm$ 0.00 | 0.57 $\pm$ 0.00 | 0.58 $\pm$ 0.01 |
> |  | ELBO | 40.53 $\pm$ 2.03 | 42.86 $\pm$ 0.95 | 36.98 $\pm$ 1.40 | 40.21 $\pm$ 2.52 |
> |  | EUBO | 58.34 $\pm$ 0.62 | 60.14 $\pm$ 0.67 | 59.20 $\pm$ 0.53 | 60.38 $\pm$ 2.90 |
> | **6** | Modes$\uparrow$ | 31.40 $\pm$ 5.03 | **32.00 $\pm$ 4.06** | **48.20 $\pm$ 2.39** | 47.80 $\pm$ 1.92 |
> |  | Spearman | 0.55 $\pm$ 0.00 | 0.55 $\pm$ 0.01 | 0.56 $\pm$ 0.00 | 0.56 $\pm$ 0.00 |
> |  | ELBO | 34.60 $\pm$ 1.87 | 35.84 $\pm$ 1.26 | 33.00 $\pm$ 1.69 | 34.73 $\pm$ 0.74 |
> |  | EUBO | 51.28 $\pm$ 0.87 | 52.58 $\pm$ 0.86 | 50.34 $\pm$ 0.72 | 50.73 $\pm$ 0.29 |
> | **8** | Modes$\uparrow$ | 59.80 $\pm$ 0.45 | **60.00 $\pm$ 0.00** | 60.00 $\pm$ 0.00 | 60.00 $\pm$ 0.00 |
> |  | Spearman | 0.73 $\pm$ 0.01 | 0.73 $\pm$ 0.01 | 0.76 $\pm$ 0.01 | 0.77 $\pm$ 0.00 |
> |  | ELBO | 27.07 $\pm$ 1.47 | 28.49 $\pm$ 2.00 | 25.85 $\pm$ 1.00 | 26.07 $\pm$ 2.09 |
> |  | EUBO | 39.76 $\pm$ 1.01 | 40.73 $\pm$ 0.48 | 40.81 $\pm$ 0.64 | 41.06 $\pm$ 0.82 |
> | **10** | Modes$\uparrow$ | 20.80 $\pm$ 5.02 | **23.40 $\pm$ 0.55** | 35.20 $\pm$ 2.28 |**37.40 $\pm$ 2.79** |
> |  | Spearman | 0.57 $\pm$ 0.00 | 0.57 $\pm$ 0.00 | 0.57 $\pm$ 0.00 | 0.57 $\pm$ 0.00 |
> |  | ELBO | 23.77 $\pm$ 2.25 | 24.37 $\pm$ 2.09 | 22.02 $\pm$ 0.83 | 23.57 $\pm$ 1.33 |
> |  | EUBO | 38.61 $\pm$ 0.73 | 39.07 $\pm$ 1.19 | 37.86 $\pm$ 0.41 | 38.94 $\pm$ 0.87 |
>
>
> References
>
> [1] Pan, Ling, et al. "Better training of gflownets with local credit and incomplete trajectories." International Conference on Machine Learning. PMLR, 2023.
>
> [2] Jang, Hyosoon, et al. “Learning Energy Decompositions for Partial Inference in GFlowNets.” The Twelfth International Conference on Learning Representations, 2024.
>
> > **[W10]** **Mistakes in formulations.** The equation on line 964 contains an error in the term $\frac{1}{2}P_B(s_{k+i-1|s_{k+i}})$.
> Additionally, there is a typographical error in Formulation (21) on line 1039, where $P_{(\alpha)}$ should be corrected to $P_{\alpha}$.
>
> Thank you for pointing this out! We've revised the paper accordingly.
>
> > **[W11]** Mismatch with previous works. Why lower number of modes for the baselines in Molecule Generation?
>
> We conduct experiments on Molecule Generation based on the source code of [1]. We use the defaulted settings in the code for all the objectives, and set the reward exponent to 4, the random action probability for $\epsilon$-greedy to 0.1 to align with the defaulted setting of [2] and to ensure fair comparisons with forward-looking objectives. In particular, we notice that the defaulted multi-process molecule rollout implementation not only potentially produces more molecules for evaluation than the batch size reported in [1,2], but also causes reproduction troubles due to the stochasticity. Hence, we follow [3] to use the single-process implementation, and modify the Message Passing mechanism following [4] to enhance the reproducibility of our results. For more details, please refer to Appendix D.1, the paper's source code in supplementary materials and the source codes of [1,2,3,4]. We are very open and willing to help you to reproduce results in the paper.
>
> References
>
> [1] Tiapkin, Daniil, et al. "Generative flow networks as entropy-regularized rl." International Conference on Artificial Intelligence and Statistics. PMLR, 2024.
>
> [2] Pan, Ling, et al. "Better training of gflownets with local credit and incomplete trajectories." International Conference on Machine Learning. PMLR, 2023.
>
> [3] Jang, Hyosoon, et al. “Learning Energy Decompositions for Partial Inference in GFlowNets.” The Twelfth International Conference on Learning Representations, 2024.
>
> [4] Yunchong, Song, et al . Ordered GNN: Ordering message passing to deal with heterophily and over-smoothing. In The Eleventh International Conference on Learning
> Representations, 2023.

---

### Official Review · Reviewer_5Rz4 · 2025-10-28

**Soundness:** 3
**Presentation:** 2
**Contribution:** 3
**Rating:** 4
**Confidence:** 3

**Summary:**

The paper proposes $\alpha$-GFlowNets, which generalizes standard GFlowNets by introducing a mixing coefficient $\alpha$ to balance exploration and exploitation. By replacing the fixed equally mixed policy ($\alpha=0.5$) with a flexible $\alpha$-weighted mixture of forward and backward policies, the method offers a flexible way to control exploration-exploitation in online training of GFlowNets. The approach shows consistent performance gains across various generative tasks, such as set, sequence, and molecule generation.

**Strengths:**

- The paper is well-written and easy to follow, with clear motivation and intuitive explanations.

- The proposed method is easy and straightforward to implement for exploration–exploitation control.

- It is insightful and interesting that the mixing coefficient $\alpha$ is introduced from the connection between Markov chain reversibility and the GFlowNet formulation.

- The proposed method demonstrates strong empirical performance across diverse benchmarks, showing improved exploration and sample diversity.

**Weaknesses:**

- My first concern stems from the ambiguity in proving equally mixed policy:
$P_{\text{0.5}} = \frac{1}{2} P_F + \frac{1}{2} P_B$.
The authors state that if $P_F(s'|s) > 0$, then $P_F(s|s') = 0$ in Line 967, as $P_F$ is one-directional. Here, it seems that the authors interpret $P_F(s|s')$ as $P_F(s_{t+1} = s ,|, s_t = s')$. However, to me, it seems that the equation should instead be written as $P_F(s|s') = P_F(s_t = s ,|, s_{t+1} = s')$, which is non-zero, since
$P_F(s_t = s ,|, s_{t+1} = s') = P_F(s_{t+1} = s' ,|, s_t = s) \frac{P_F(s_t = s)}{P_F(s_{t+1} = s')}$
(I think this misunderstanding might stem from time conditioning). Could the authors further clarify this point?

- How does the role of $\alpha$ differ from temperature-scaling of rewards in terms of exploration–exploitation? Furthermore, why do the authors not compare it with the temperature-scaling method? Although the authors derive $\alpha$ from a different perspective, its role seems the same as that of the temperature parameter. Could the authors further clarify this point?

**Questions:**

See weaknesses.

---

> ### Author Response · Authors · 2025-11-28
> **Response to Reviewer 5Rz4 (1/3)**
>
> We thank the reviewer for recognizing the theoretical novelty of α‑GFN, its simple yet effective design, the improved empirical performance, and the readability of the paper. Below, we address each of the reviewer’s concerns point by point.
>
> > **[W1]** Mistakes in the formulation of $P_F(s|s')$ as $P_F(s|s')=P_F(s_{t+1}=s | s_{t}=s')$ and the one-directional property? Why $P_F(s|s')=0$ if $P_F(s'|s)>0$?
>
> - First, we clarify the **Markovian property** of the forward policy $P_F(\cdot | \cdot): \mathbb{A} \rightarrow \mathbb{R}_+$. It is a time-homogeneous Markov transition kernel [1,2], which possesses two key properties:
>     - It travels from any time $t$ to the corresponding next time $t+1$ by definition of Markov kernels (**note that the time indices here cannot be swapped**).
>     - Its value is independent of the visiting time. Given any $(s,s') \in \mathbb{A}$, the value of $P_F(s|s')$ is independent of the time when $(s,s')$ is visited. That is, $P_F(s_{t+1}=s|s_t=s') \equiv P_F(s|s')$ for any time $t$, which is why we use the notation $P_F(s|s')$ in the paper.
> - Second, we clarify the **one-directional property** of $P_F$. The policy $\tilde{P_F}(s_t=s | s_{t+1}=s')=P_F(s_{t+1}=s'|s_{t}=s) \frac{P_F(s_t=s)}{P_F(s_{t+1}=s')}$ is the time-reversal of $P_F$. It is a policy different from $P_F$. The equality $\tilde{P_F}(s_t=s | s_{t+1}=s')=P_F(s|s')$ generally fails unless $P_F$ itself satisfies time-reversibility (which is equivalent to satisfying conditions such as detailed balance or Kolmogorov's criterion). Unfortunately, it is not the case in GFlowNets. $P_F$ is a GFlowNet policy whose transitions form a directed acyclic graph (DAG). This means $P_F(s|s')$ must be 0 if $P_F(s'|s)>0$, otherwise there would be loops within the DAG.
>
> References
>
> [1] Yoshua Bengio, Salem Lahlou, Tristan Deleu, Edward J Hu, Mo Tiwari, and Emmanuel Bengio. GFlowNet foundations. Journal of Machine Learning Research, 24(210):1–55, 2023.
>
> [2] Tristan Deleu and Yoshua Bengio. Generative flow networks: a Markov chain perspective. arXiv preprint arXiv:2307.01422, 2023.

---

> ### Author Response · Authors · 2025-11-28
> **Response to Reviewer 5Rz4 (2/3)**
>
> > **[W2]** How does $\alpha$-GFN differ from temperature-scaling of rewards in terms of exploration-exploitation? Why no comparison with temperature-scaling?
>
> Thank you for the thoughtful question. We illustrate the difference between temperature-scaling and $\alpha$-GFN from both theoretical and the empirical viewpoints.
> - Theoretically,
>   - **temperature scaling of reward preserves the flow balancing structure of GFlowNets**. However, **$\alpha$-GFN suggests a systematic way to violate the flow structure during training** via its link to the Markov chain theories. This violation of flows not only leads to performance improvements and distinguishes $\alpha$-GFN from other approaches, but more importantly, **reveals a deeper connection between GFlowNets and Markov chain theories** (see Appendix B). In other words, apart from empirical results, this paper further bridges the theoretical frameworks between GFlowNets and Markov chains, and shows one benefit of such a closer connection: a flexible exploration-exploitation trade-off by tuning a flow imbalancing term $\alpha$.
>
>   - In terms of loss formulations, **we find the forward-looking variants [1] to be more similar to $\alpha$-GFNs than reward temperature scaling, since forward-looking losses inject intermediate energies into each transition**, and temperature-scaling of reward does not add any term into the intermediate transitions. Therefore, we test $\alpha$-GFN on the forward-looking objectives (FL-DB in Table 1, FL-DB and FL-SubTB($\lambda$) in Table 3) and find significant performance gains (~1000% more modes in the large-set setting of Table 1, 177% more modes for FL-DB and 145% more modes for FL-SubTB($\lambda$)). These results suggest that $\alpha$-GFN is fundamentally different from previous approaches.
>
> - Empirically, **$\alpha$-GFN and temperature scaling of reward are distinct and can be complementary in some scenarios**. The two approaches can be further combined under careful tuning of both $\alpha$ and the reward temperature. To be specific, we carry out a case study on how $\alpha$-GFN performs under the default temperature, a low reward temperature ($0.5\times$ the default temperature), and a high reward temperature ($2\times$ the default temperature) regimes in Set Generation. We directly reuse the best $\alpha$ under the default temperature (which produces the results in Table 1 and Table 3), and show the results in what follows. All experiments reported below are run with 5 seeds.
>
> **Table 1, Set Generation, DB objective, Temperature-scaling of Reward**
> | Set Size | Reward Temperature | Low |  | Default |  | High |  |
> |---------|---------------------|------|------|---------|------|--------|------|
> |         | **Metric/Objective** | Baseline | Ours | Baseline | Ours | Baseline | Ours |****
> | **Small**  | Modes$\uparrow$  | 89.60 $\pm$ 0.55 | **90.00 $\pm$ 0.00**  | 24.80 $\pm$ 2.59 | **55.80 $\pm$ 13.92**  | 5.60 $\pm$ 2.07 | **13.20 $\pm$ 6.46**  |
> |  | Top-1000 Reward$\uparrow$  | 0.22 $\pm$ 0.00 | 0.22 $\pm$ 0.00  | 0.18 $\pm$ 0.00 | **0.20 $\pm$ 0.01**  | 0.13 $\pm$ 0.00 | **0.16 $\pm$ 0.01**  |
> |  | Spearman Corr  | 1.00 $\pm$ 0.00 | 1.00 $\pm$ 0.00  | 1.00 $\pm$ 0.00 | 1.00 $\pm$ 0.00  | 0.99 $\pm$ 0.00 | 0.99 $\pm$ 0.00  |
> | **Medium**  | Modes$\uparrow$  | 20.40 $\pm$ 6.23 | **414.14 $\pm$ 159.54**  | 0.00 $\pm$ 0.00 | **31.40 $\pm$ 34.06**  | 0.00 $\pm$ 0.00 | **6.00 $\pm$ 9.03**  |
> |  | Top-1000 Reward$\uparrow$  | 530369.78 $\pm$ 11276.62 | **703830.10 $\pm$ 19760.58**  | 110427.38 $\pm$ 5337.25 | **552020.09 $\pm$ 59210.23**  | 10866.33 $\pm$ 590.79 | **468720.01 $\pm$ 61572.59**  |
> |  | Spearman Corr  | 1.00 $\pm$ 0.00 | 0.99 $\pm$ 0.00  | 0.99 $\pm$ 0.00 | 0.97 $\pm$ 0.01  | 0.98 $\pm$ 0.01 | 0.93 $\pm$ 0.01  |
> | **Large**  | Modes$\uparrow$  | 35.40 $\pm$ 9.21 | **8727.00 $\pm$ 3678.32**  | 0.00 $\pm$ 0.00 | **355.80 $\pm$ 319.84**  | 0.00 $\pm$ 0.00 | **75.40 $\pm$ 139.28**  |
> |  | Top-1000 Reward$\uparrow$  | 437695.33 $\pm$ 11484.61 | **868599.17 $\pm$ 9984.98**  | 58086.54 $\pm$ 5216.95 | **624721.77 $\pm$ 55336.88**  | 4367.90 $\pm$ 262.49 | **448232.23 $\pm$ 112762.17**  |
> |  | Spearman Corr  | 0.99 $\pm$ 0.00 | 0.97 $\pm$ 0.01  | 0.98 $\pm$ 0.00 | 0.94 $\pm$ 0.00  | 0.96 $\pm$ 0.00 | 0.84 $\pm$ 0.04  |

---

> ### Author Response · Authors · 2025-11-28
> **Response to Reviewer 5Rz4 (3/3)**
>
> **Table 2, Set Generation, FL-DB objective, Temperature-scaling of Reward**
> | Set Size | Reward Temperature | Low |  | Default |  | High |  |
> |---------|---------------------|------|------|---------|------|--------|------|
> |         | **Metric/Objective** | Baseline | Ours | Baseline | Ours | Baseline | Ours |
> | **Small**  | Modes$\uparrow$  | 90.00 $\pm$ 0.00 | 90.00 $\pm$ 0.00  | 83.00 $\pm$ 2.45         | **88.60 $\pm$ 2.19** | 28.80 $\pm$ 4.82 | **74.20 $\pm$ 12.81**  |
> |  | Top-1000 Reward$\uparrow$  | 0.22 $\pm$ 0.00 | 0.22 $\pm$ 0.00  | 0.22 $\pm$ 0.00| 0.22 $\pm$ 0.00          | 0.19 $\pm$ 0.01|**0.21 $\pm$ 0.01**|
> |  | Spearman Corr  | 0.94 $\pm$ 0.01 | 0.83 $\pm$ 0.02  | 0.99 $\pm$ 0.00          | 0.97 $\pm$ 0.01          | 0.99 $\pm$ 0.01 | 0.99 $\pm$ 0.00  |
> | **Medium**  | Modes$\uparrow$  | 1254.20 $\pm$ 398.33 | **8125.00 $\pm$ 1435.04**  | 14.20 $\pm$ 10.06 | **118.60 $\pm$ 45.88** | 0.40 $\pm$ 0.89 | **4.40 $\pm$ 3.58**  |
> |  | Top-1000 Reward$\uparrow$  | 762543.02 $\pm$ 18174.10 | **875253.40 $\pm$ 14126.35**  | 507462.73 $\pm$ 47471.45 | **638688.10 $\pm$ 24711.41** | 254175.71 $\pm$ 74676.34 | **458420.14 $\pm$ 40338.86**  |
> |  | Spearman Corr  | 0.88 $\pm$ 0.02 | 0.74 $\pm$ 0.02  | 0.97 $\pm$ 0.01          | 0.93 $\pm$ 0.01          | 0.95 $\pm$ 0.02 | 0.99 $\pm$ 0.00  |
> | **Large**  | Modes$\uparrow$  | 12932.20 $\pm$ 1297.15 | **19088.00 $\pm$ 1377.06**  | 247.60 $\pm$ 147.17| **2239.20 $\pm$ 169.80**     | 0.80 $\pm$ 0.84 | **57.40 $\pm$ 27.57**  |
> |  | Top-1000 Reward$\uparrow$  | 874968.47 $\pm$ 2683.06 | **875841.37 $\pm$ 2985.08**  | 593090.24 $\pm$ 67865.24 | **768544.60 $\pm$ 6904.37**  | 177285.24 $\pm$ 44777.47 | **472091.19 $\pm$ 50150.98**  |
> |  | Spearman Corr  | 0.80 $\pm$ 0.01 | 0.72 $\pm$ 0.02  | 0.90 $\pm$ 0.02          | 0.85 $\pm$ 0.02          | 0.89 $\pm$ 0.02 | 0.95 $\pm$ 0.02  |
>
>
> **Table 3, Set Generation, TB objective, Temperature-scaling of Reward**
>
> | Set Size | Reward Temperature | Low |  | Default |  | High |  |
> |---------|---------------------|------|------|---------|------|--------|------|
> |         | **Metric/Objective** | Baseline | Ours | Baseline | Ours | Baseline | Ours |
> | **Small**  | Modes$\uparrow$  | 86.60 $\pm$ 2.30 | **90.00 $\pm$ 0.00**  | 23.20 $\pm$ 2.39 | **24.80 $\pm$ 3.27**  | 5.00 $\pm$ 2.83 | **5.20 $\pm$ 2.77**  |
> |  | Top-1000 Reward$\uparrow$  | 0.22 $\pm$ 0.00 | 0.22 $\pm$ 0.00  | 0.18 $\pm$ 0.00 | 0.18 $\pm$ 0.00  | 0.13 $\pm$ 0.00 | 0.13 $\pm$ 0.00  |
> |  | Spearman Corr  | 1.00 $\pm$ 0.00 | 1.00 $\pm$ 0.00  | 1.00 $\pm$ 0.00 | 1.00 $\pm$ 0.00  | 1.00 $\pm$ 0.00 | 1.00 $\pm$ 0.00  |
> | **Medium**  | Modes$\uparrow$  | 1.20 $\pm$ 0.84 | **3861.40 $\pm$ 796.46**  | 0.00 $\pm$ 0.00 | **499.00 $\pm$ 427.07**  | 0.00 $\pm$ 0.00 | **16.00 $\pm$ 20.53**  |
> |  | Top-1000 Reward$\uparrow$  | 274631.96 $\pm$ 2667.57 | **821655.62 $\pm$ 12654.71**  | 44854.41 $\pm$ 1480.97 | **703904.36 $\pm$ 39194.70**  | 7846.57 $\pm$ 388.28 | **476281.30 $\pm$ 128781.91**  |
> |  | Spearman Corr  | 1.00 $\pm$ 0.00 | 0.99 $\pm$ 0.00  | 1.00 $\pm$ 0.00 | 0.86 $\pm$ 0.30  | 1.00 $\pm$ 0.00 | 0.99 $\pm$ 0.00  |
> | **Large**  | Modes$\uparrow$  | 0.20 $\pm$ 0.45 | **58.00 $\pm$ 8.06**  | 0.00 $\pm$ 0.00 | **591.60 $\pm$ 210.56**  | 0.00 $\pm$ 0.00 | **0.40 $\pm$ 0.55**  |
> |  | Top-1000 Reward$\uparrow$  | 102171.52 $\pm$ 1369.26 | **481987.84 $\pm$ 8520.90**  | 11754.28 $\pm$ 230.91 | **683702.46 $\pm$ 32706.82**  | 1663.17 $\pm$ 29.14 | **191334.11 $\pm$ 35994.30**  |
> |  | Spearman Corr  | 0.99 $\pm$ 0.00 | 0.99 $\pm$ 0.00  | 1.00 $\pm$ 0.00 | 1.00 $\pm$ 0.00  | 1.00 $\pm$ 0.00 | 1.00 $\pm$ 0.00  |
>
> References
>
> [1] Pan, Ling, et al. "Better training of gflownets with local credit and incomplete trajectories." International Conference on Machine Learning. PMLR, 2023.

---

### Official Review · Reviewer_qdTi · 2025-11-01

**Soundness:** 2
**Presentation:** 4
**Contribution:** 3
**Rating:** 6
**Confidence:** 4

**Summary:**

The paper proposes $\alpha$-GFN, a generalization of GFlowNet conventional training, where forward and backward policies are mixed with a tunable coefficient \(\alpha\in(0,1)\). To derive the $\alpha$-GFN, the author first connects standard GFlowNet objectives (DB/SubTB/TB) to sub-trajectory reversibility of Markovian chains under the equally mixed kernel $P_{0.5}=\tfrac12 P_F+\tfrac12 P_B$; Then they extends this to $P_\alpha=\alpha P_F+(1-\alpha)P_B$, leading to the derivation of $\alpha$-GFN objectives that admit unique flow solutions. The paper further introduces a simple two-stage $\alpha$-schedule that anneals toward 0.5, thereby recovering the standard GFlowNet target at convergence. Experiments on set generation, bit-sequence generation, and a molecule generation demonstrate improvements and suggest that  $\alpha$ serves as an interpretable exploration–exploitation knob.

**Strengths:**

* The paper provides an  advanced elucidation of the connections between MCMC sampling and GFlowNets.

* The mapping from DB/SubTB/TB balance conditions to reversibility under $P_{0.5}$, and the extension to $P_\alpha$ is conceptually neat and well-motivated.

* The $\alpha$-mixing idea applies across canonical objectives (DB/SubTB/TB) and forward-looking variants with minimal code changes.

* The gradient perspective clarifies how $\alpha$ biases credit assignment toward exploitative $\alpha>0.5$ or exploratory regimes $\alpha<0.5$.

* The proposed two-stage annealing schedule is simple to implement and integrates naturally into existing GFlowNet workflows.

**Weaknesses:**

* The key limitation is that while the $\alpha$-balance condition can ensure a unique flow at convegence. This flow may not be equal to the unique target flow $F^\ast(\tau)=P_B(\tau|x)R(x)$.  Although the annealing strategy that gradually pushes $\alpha$ toward $0.5$ can alleviate this issue, further theoretical or empirical study of the gaps between the unique flow under $\alpha\neq 0.5$ and the target flow $F^\ast$ corresponding to $\alpha=0.5$ would strengthen the paper’s claims.  ( I note that in Figure 3, only the choices of $\alpha$ closer to 0.5 would give good performance. )

*  In addition to reporting top-k average rewards or number of modes (used for evaluating exploration capability), please also report diversity/similarity metrics for all three set of experiments. This would make it clearer whether varying $\alpha$ indeed provides an explicit control over the exploration–exploitation trade-off.

* According to Fig.17 and 18, the performance gap between $\alpha$-GFN and vanilla GFN is trivial for the real-world benchmarks.

**Questions:**

1. What's the definition of $f$ in line 6 of Algo.1 ?

---

> ### Author Response · Authors · 2025-11-28
> **Response to Reviewer qdTi (1/3)**
>
> We thank the reviewer for recognizing the theoretical novelty of α‑GFN, the insights on its exploration–exploitation behavior, and its easy-to-implement design. Below, we address each of the reviewer’s concerns point by point.
>
> > **[W1]** In $\alpha$-GFN, the converged flows may not satisfy $F^*(\tau)=P_B(\tau|x)R(x)$ like vanilla GFlowNets. Any study of the gap of flows at $\alpha \neq 0.5$ and $\alpha = 0.5$?
>
> - From a theoretical viewpoint, **$\alpha$-GFN is an attempt to systematically imbalance the flows and violate the flow structure of GFlowNets**. That is, under $\alpha \neq 0.5$, the target flow $F^*(\tau)=P_B(\tau|x)R(x)$ may not be satisfied, which naturally leads to the drop in Spearman correlations observed in Figure 3. Nevertheless, the convergence of $\alpha$-GFN can still be guaranteed via the link to Markov chain theories. **In other words, convergence under imbalanced flows reveals a deeper connection between GFlowNets and Markov chains, as detailed in Appendix B**. Moreover, we show in Section 4.4 a benefit of such imbalance: flexible exploration-exploitation trade-offs. In short, **though $\alpha$-GFNs may not satisfy the common target of GFlowNets, they can converge and improve performance**.
>
> - **Empirically, we find that $\alpha$-GFNs find more modes than vanilla GFlowNets even though their Spearman correlations are relatively lower sometimes**. Take Figure 3 as an example, which visualizes the Spearman correlation dynamics in large sets. In terms of modes, the best $\alpha$ here for DB, FL-DB and TB are 0.9, 0.9 and 0.7 respectively, all of which have lower Spearman correlations than vanilla GFlowNets. However, their number of modes is significantly greater than vanilla GFlowNets'. As shown in Table 1, $\alpha$-GFN finds at least 1000% more modes than vanilla GFlowNets in large sets. In Bit Sequence Generation and Molecule Generation, the cases are similar: $\alpha$-GFN sometimes shows lower Spearman correlations, but a higher number of modes. **These results also suggest that Spearman correlations and the number of modes do not seem to be tightly coupled**, indicating that correlation alone may not be a reliable proxy for mode discovery. **By noticing that the goal of GFlowNets in discovery tasks is to find more distinct high-reward modes, we argue that $\alpha$-GFNs show better performance than vanilla GFlowNets** even though they do not fit the reward distribution as well as vanilla GFlowNets.

---

> ### Author Response · Authors · 2025-11-28
> **Response to Reviewer qdTi (2/3)**
>
> > **[W2]** Report diversity or similarity metrics.
>
> Thank you for your helpful comments! For the similarity/diversity in Molecule Generation, please refer to Table 3. Meanwhile, it is also worth noting that the number of modes itself is is itself an indicator of similarity/diversity since modes are distinct high-reward samples. Moreover, the maximum number of modes in Bit Sequence Generation is predetermined (60 modes are set in advance, and the policy is trained to find all 60 modes without replacement). In addition, we evaluate the Jaccard similarity of the top-1000 highest-reward sets in the Set Generation task over 5 seeds, and summarize the results as follows:
>
> **Table: diversity/similarity evaluation in Set Generation**
> | Set Size | Objective | TB |  | FL-DB |  | DB |  |
> |---------|---------------------|------|------|---------|------|--------|------|
> |         | Metric | Baseline | Ours | Baseline | Ours | Baseline | Ours |
> | **Small**  | Modes$\uparrow$  | 23.20 $\pm$ 2.39 | **24.80 $\pm$ 3.27**  | 83.00 $\pm$ 2.45 | **88.60 $\pm$ 2.19**  | 24.80 $\pm$ 2.59 | **55.80 $\pm$ 13.92**  |
> |  | Top-1000 Reward$\uparrow$  | 0.18 $\pm$ 0.00 | 0.18 $\pm$ 0.00  | 0.22 $\pm$ 0.00 | 0.22 $\pm$ 0.00  | 0.18 $\pm$ 0.00 | **0.20 $\pm$ 0.01**  |
> |  | Top-1000 Similarity  | 0.69 $\pm$ 0.00 | 0.69 $\pm$ 0.00  | **0.73 $\pm$ 0.00** | 0.74 $\pm$ 0.00  | **0.69 $\pm$ 0.00** | 0.72 $\pm$ 0.01  |
> |  | Spearman Corr  | 1.00 $\pm$ 0.00 | 1.00 $\pm$ 0.00  | 0.99 $\pm$ 0.00 | 0.97 $\pm$ 0.01  | 1.00 $\pm$ 0.00 | 1.00 $\pm$ 0.00  |
> | **Medium**  | Modes$\uparrow$  | 0.00 $\pm$ 0.00 | **499.00 $\pm$ 427.07**  | 14.20 $\pm$ 10.06 | **118.60 $\pm$ 45.88**  | 0.00 $\pm$ 0.00 | **31.40 $\pm$ 34.06**  |
> |  | Top-1000 Reward$\uparrow$  | 44854.41 $\pm$ 1480.97 | **703904.36 $\pm$ 39194.70**  | 507462.73 $\pm$ 47471.45 | **638688.10 $\pm$ 24711.41**  | 110427.38 $\pm$ 5337.25 | **552020.09 $\pm$ 59210.23**  |
> |  | Top-1000 Similarity  | **0.69 $\pm$ 0.00** | 0.86 $\pm$ 0.01  | **0.78 $\pm$ 0.01** | 0.81 $\pm$ 0.01  | **0.71 $\pm$ 0.00** | 0.80 $\pm$ 0.01  |
> |  | Spearman Corr  | 1.00 $\pm$ 0.00 | 0.86 $\pm$ 0.30  | 0.97 $\pm$ 0.01 | 0.93 $\pm$ 0.01  | 0.99 $\pm$ 0.00 | 0.97 $\pm$ 0.01  |
> | **Large**  | Modes$\uparrow$  | 0.00 $\pm$ 0.00 | **591.60 $\pm$ 210.56**  | 247.60 $\pm$ 147.17 | **2239.20 $\pm$ 169.80**  | 0.00 $\pm$ 0.00 | **355.80 $\pm$ 319.84**  |
> |  | Top-1000 Reward$\uparrow$  | 11754.28 $\pm$ 230.91 | **683702.46 $\pm$ 32706.82**  | 593090.24 $\pm$ 67865.24 | **768544.60 $\pm$ 6904.37**  | 58086.54 $\pm$ 5216.95 | **624721.77 $\pm$ 55336.88**  |
> |  | Top-1000 Similarity  | **0.74 $\pm$ 0.00** | 0.87 $\pm$ 0.01  | **0.85 $\pm$ 0.01** | 0.87 $\pm$ 0.00  | **0.76 $\pm$ 0.00** | 0.88 $\pm$ 0.01  |
> |  | Spearman Corr  | 1.00 $\pm$ 0.00 | 1.00 $\pm$ 0.00  | 0.90 $\pm$ 0.02 | 0.85 $\pm$ 0.02  | 0.98 $\pm$ 0.00 | 0.94 $\pm$ 0.00  |

---

> ### Author Response · Authors · 2025-11-28
> **Response to Reviewer qdTi (3/3)**
>
> > **[W3]** According to Fig.17 and 18, the performance gap between $\alpha$-GFN and vanilla GFN is trivial for the real-world benchmarks.
>
> We respectfully disagree that the improvement is trivial. **While the gap in Figures 17 and 18 may appear modest on some metrics, Figure 15 shows that $\alpha$‑GFN achieves substantially better mode discovery on the same real‑world benchmark, Molecule Generation**. The results in the three figures are detailed in Table 3, which we present in the following:
>
> **Table 1: Molecule Generation results (reproduced from Table 3 of the paper)**
>
> |Objective|DB||FL-DB||SubTB(λ)||FL-SubTB(λ)||TB||
> |-|-|-|-|-|-|-|-|-|-|-|
> |Metric|Baseline|Ours|Baseline|Ours|Baseline|Ours|Baseline|Ours|Baseline|Ours|
> | Modes↑|10.00 ± 3.08|**14.40 ± 2.41**|9.00 ± 1.58|**25.00 ± 4.36**|22.20 ± 2.77|**26.40±2.30**|16.00 ± 3.16|**39.20±7.05**|38.40 ± 6.11|**40.20 ± 5.07**|
> |Top-100 Reward↑|6.81 ± 0.03|**6.84 ± 0.03**|6.97±0.02|**7.03±0.05**|6.97 ± 0.03|**6.99±0.02**|6.98±0.06|**7.12±0.04**|7.14 ± 0.05|**7.16±0.05**|
> |Top-100 Similarity↓|0.57±0.01|**0.56 ± 0.01**|0.74±0.05|**0.51±0.03**|0.57±0.01|0.58±0.02|**0.49±0.05**|0.57±0.02|**0.63±0.03**|0.64±0.04|
>
> As can be seen in the table above and Figure 17, even though metrics such as Top-100 reward and Top-100 similarity exhibit some fluctuations, $\alpha$‑GFN consistently discovers substantially more modes than the vanilla GFlowNet, meeting the primary target of GFlowNets in discovery tasks.
>
> Moreover, **we additionally study $\alpha$-GFN in another real-world task, LLM reasoning, FlowRL[1]**. To be specific, we apply $\alpha$-GFN with α ∈ {0.1, 0.5, 0.9} to the objective of FlowRL without any scheduling (fixing $\alpha$=$\alpha_0$ throughout the whole training process). Then, we use the same training/evaluation strategy as FlowRL on the math domain with Qwen2.5-3B-Instruct and the official implementation of FlowRL in the VeRL [2] recipe for 200 steps (the maximum response length is set to be 8k). We report the performance on mathematical reasoning tasks as per FlowRL in the following:
>
> **Table 2: performance of different $\alpha$ values in FlowRL**
> | Benchmark                    | $\alpha$ = 0.1 | $\alpha$ = 0.5 | $\alpha$ = 0.9 |
> |---------------------------|------------:|------------:|------------:|
> | AIME2024       |   0.054167  |   **0.056250**  |   **0.056250**  |
> | AIME2025       |   0.033333  |   0.031250  |   **0.045833**  |
> | AMC23          |   0.385937  |   **0.542188**  |   0.496875  |
> | Math500        |   **0.618875**  |   0.587375  |   0.547500  |
> | Minerva        |   **0.240119**  |   0.232537  |   0.186351  |
> | Olympiabench   |   **0.273090**  |   0.243694  |   0.231825  |
>
> A clear trend is observed in Math500, Minerva and Olympiabench: as $\alpha$ decreases, the model benefits from better exploration. In challenging tasks like AIME2024 and AIME2025, enhanced exploitation with larger $\alpha$ values helps. AMC23 lies between these two regimes: the best performance is achieved at the intermediate value $\alpha=0.5$. **These results show that the exploration-exploitation trade-offs with $\alpha$ scale up to various domains**.
>
> To better understand how $\alpha$ works in this case, we investigate the training metrics at step 200 with a focus on the two: (1)average reward and (2) the KL divergence between the trained policy and the reference policy (termed `ref_kl`), i.e. the average of $\pi_{ref}-\pi_{\theta}$, and summarize them in what follows:
>
> **Table 3: exploration–exploitation indicators (step 200, FlowRL)**
> |Metric|$\alpha$ = 0.1|$\alpha$ = 0.5|$\alpha$ = 0.9|
> |-|-|-|-|
> |Average Reward|**-0.506**|-0.399|-0.431|
> |`ref_kl`|-0.298|-0.2111|**-0.680**|
>
> It can be seen above that $\alpha=0.1$ recieves relatively low reward due to the exploration, while the policy trained with $\alpha=0.9$ is significantly larger than its reference policy and also achieves higher reward than $\alpha=0.1$ due to the exploitation. These results further validate the analysis of exploration-exploitation trade-offs in Section 4.4.
>
> References
> [1] Zhu, Xuekai, et al. "Flowrl: Matching reward distributions for llm reasoning." arXiv preprint arXiv:2509.15207 (2025).
> [2] Sheng, Guangming, et al. "Hybridflow: A flexible and efficient rlhf framework." Proceedings of the Twentieth European Conference on Computer Systems. 2025.
>
> > **[Q1]** What's the definition of $f$ in line 6 of Algo.1 ?
>
> Thank you for pointing this out! $f$ is a scheduling function, and we use in all the experiments an exponential annealing $f$. Given the total number of steps $N$, stage 1 steps $N_1$, the initial $\alpha_0$, $f$ is defined as
> $$
> f(\alpha_0,n,N_1,N)=0.5+(\alpha_0-0.5)\exp (-4 \cdot \frac{n-N_1}{N-N_1})
> $$
> We've also revised the paper accordingly. Thank you again for this helpful comment!

---

### Official Review · Reviewer_UYFs · 2025-11-01

**Soundness:** 3
**Presentation:** 2
**Contribution:** 3
**Rating:** 4
**Confidence:** 4

**Summary:**

This paper presents $\alpha$-GFN, a novel and well-motivated generalization of GFlowNets that introduces a hyperparameter $\alpha$ to control the weighting between the forward and backward policies in the training objective by viewing GFN as an MDP with transition $P=\alpha P_F+(1-\alpha)P_B$. The authors derive the corresponding novel training objectives, and scheduling strategies of $\alpha$, leading to better exploration-exploitation trade-off and hence significant performance gains in discovering diverse, high-reward modes across several tasks.

**Strengths:**

The core idea of inbalanced mixing is a novel and well-motivated generalization of GFN. Both the theoretical foundation and the empirical results are solid. The analysis of the exploration-exploitation behavior and $\alpha$ is insightful.

**Weaknesses:**

I feel that the authors leave too many required theoretical analysis to the appendix, making the main body somehow hard to follow.

**Questions:**

1. When $\alpha\neq0.5$, does the terminating distribution induced from the forward policy $P_F$ match with the reward $R$? This is an essential question that motivates the design of $\alpha$-scheduling, which I think the authors should clarify in earlier sections.
2. In appendix B.4, the authors assume $P_B$ to have the same stationary distribution as $P$. Can the authors explain the reason for this assumption further?

---

> ### Author Response · Authors · 2025-11-28
> **Response to Reviewer UYFs**
>
> We thank the reviewer for recognizing the theoretical novelty of α‑GFN and our analysis of its exploration–exploitation behavior. Below, we address each of the reviewer’s concerns point by point.
>
> > **[W1]** I feel that the authors leave too many required theoretical analysis to the appendix, making the main body somehow hard to follow.
>
> We appreciate the reviewer’s concern about placing much of the theoretical analysis in the appendix. Due to space limitations, it is unfortunately not feasible to include the full technical development in the main text, so we opted to keep only the key intuitions and statements in the main body and move detailed theorems and proofs to the appendix. We would be happy to incorporate any specific suggestions into the camera-ready version.
>
> > **[Q1]** Does the terminating distribution match with the reward when $\alpha \neq 0.5$?
>
> - First, when $\alpha \neq 0.5$, the flows are imbalanced, and the flow structure of GFlowNets is violated to some extent. Hence, **the terminating distribution may not match the reward as well as vanilla GFlowNets**. As shown in Figure 3, tuning $\alpha$ may drive the policy away from the reward distribution, leading to a drop in the correlation metrics. However, the convergence of $\alpha$-GFN is still ensured due to the link to Markov chain theories, though the target of such convergence may not be the same as vanilla GFlowNets. Moreover, the mode discovery ability can be significantly enhanced under the flow imbalance even though correlation metrics drop sometimes (see Tables 1-3 and Figure 3).
>
> - Next, we detail the motivations behind the design of $\alpha$-scheduling.
>     - We find empirically that **the imbalance of flows introduced by $\alpha$-GFN brings performance improvements in terms of modes even without scheduling**. For instance, we show in Figure 4(a) that at stage 1 ($\alpha$ is kept at its initial value), $\alpha$-GFN already consistently discovers more modes than vanilla GFlowNets in Molecule Generation. In Figure 8, it is also shown that $\alpha$-GFN already achieves performance gains at stage 1 in Set Generation.
>     - We also find that **correlation metrics and the number of modes may not be tightly coupled**. Take Figure 3 as an example, which visualizes the Spearman correlation dynamics in large sets. In terms of modes, the best $\alpha$ here for DB, FL-DB and TB are 0.9, 0.9 and 0.7 respectively, all of which have lower Spearman correlations than vanilla GFlowNets. However, their number of modes are significantly greater than vanilla GFlowNets'. As shown in Table 1, $\alpha$-GFN finds at least 1000% more modes than vanilla GFlowNets in large sets. In other words, $\alpha$-GFN can discover more modes, though in this case the forward policy may mismatch with the reward.
>     - We attempt to design an $\alpha$-scheduling strategy to combine the mode discovery ability of $\alpha \neq 0.5$, a result of flexible exploration-exploitation trade-offs, and the reward-fitting ability of $\alpha=0.5$. Hence, in Algorithm 1, we first use $\alpha \neq 0.5$ to discover more modes, then use an annealing strategy at stage 2 to recover the forward policy's reward-fitting ability.
>
> > **[Q2]** Why assume $P_B$ to have the same stationary distribution as $P$ in Appendix B.4?
>
> Thank you for pointing this out! This assumption was made following the detailed balance in vanilla GFlowNets, where the forward and backward policies are trained to be the time-reversal of each other and share the same stationary distribution. Nevertheless, we find that such an assumption may be too strong in $\alpha$-GFNs, and we've revised the paper accordingly by removing Appendix B.4 for theoretical rigor.

---

### Author Response · Authors · 2025-12-02
**Summary of Rebuttal for New Area Chair**

Dear Area Chair,

Thank you for overseeing the evaluation of our submission and for the reviewers’ thoughtful feedback. To support a rapid understanding of the paper and our rebuttal, we briefly summarize the key points below.

## Key strengths highlighted by reviewers

1. **Strong theoretical innovations to bridge GFlowNets and Markov chains** (@Reviewers UYFs, qdTi, 5Rz4, Cy2F). Reviewers highlight that the paper provides a substansive connection between standard GFlowNets and Markov chains, offering a new theoretical perspective on GFlowNets.
2. **Well-motivated and easy-to-implement design of $\alpha$-GFN** (@Reviewers UYFs, qdTi, 5Rz4, Cy2F). Reviewers agree that $\alpha$-GFN is well-motivated by the theoretical connections and, together with the $\alpha$-scheduling algorithm (Algorithm 1), is easy to integrate into existing GFlowNet objectives with minimal code changes.

3. **Insightful exploration–exploitation analysis via gradients** (@Reviewers UYFs, qdTi). Reviewers appreciate the gradient‑based analysis that explains how α biases credit assignment and induces different exploration–exploitation regimes.
4. **Strong empirical performance** (@Reviewer 5Rz4). One reviewer notes that the proposed method demonstrates strong empirical performance across varied benchmarks, showing improved exploration and sample diversity.
5. **The secondary finding of trajectory-length control** (@Reviewer Cy2F). One reviewer notes that $\alpha$ indirectly controls trajectory length, offering new insight for length‑controllable GFlowNet sampling and suggesting a potential future research direction.

## Main concerns and our responses
1. **Justification of the imbalanced-flow design**. We clarify that $\alpha$-GFN is explicitly designed to introduce flow imbalance, so the converged distribution may not exactly match the reward distribution, but convergence is still guaranteed via the Markov chain connection and empirically improves mode discovery, which motivates our two-stage $\alpha$-schedule. (c.f. Reviewer UYFs, Q1)

2. **Shared stationary distribution assumption**. We acknowledge that assuming $P_B$ shares the same stationary distribution as $P$ is too strong in the $\alpha$-GFN setting and remove the corresponding part (Appendix B.4) for better theoretical rigor. (c.f. Reviewer UYFs, Q2)

3. **Difference from reward temperature scaling**. We distinguish $\alpha$-GFN from reward temperature scaling by showing that temperature scaling preserves balanced flows while $\alpha$-GFN systematically violates flow balance, and we back this up empirically on Set Generation with DB/FL-DB/TB across all set sizes, where $\alpha$-GFN remains effective across all three temperature settings (low, defaulted, high). (c.f. Reviewer 5Rz4, W2)

4. **Additional evaluation metrics**. We add similarity/diversity measures (e.g., Jaccard similarity of top-reward sets, c.f. Reviewer qdTi, W2) and ELBO/EUBO metrics (c.f. Reviewer Cy2F, W8) to further demonstrate the performance.

5. **More baselines and objectives**. We incorporate additional baselines and settings， including QGFN, Adaptive Teachers (c.f. Reviewer Cy2F, W1), SubTB/FL-SubTB on small/medium sets, FL-DB/FL-SubTB on Bit Sequence with $k=4,6,8,10$ (c.f. Reviewer Cy2F, W9), and observe that adding $\alpha$-GFN consistently improves performance.

6. **Real-world performance**. We further evaluate $\alpha$-GFN on LLM reasoning tasks following FlowRL and show that varying $\alpha$ induces the expected exploration–exploitation behavior and brings performance gains on several math benchmarks. (c.f. Reviewer qdTi, W3)


We have revised the paper accordingly, and we are grateful to all reviewers and to you for the careful evaluation of our work. We hope that, with the clarified theory, additional baselines/evaluation metrics, and new experiments, the paper will be viewed as a solid and timely contribution to the GFlowNet literature.

---

### Meta-Review · Area_Chair_FM7W · 2026-01-05

**Summary:**

Strength: The paper proposes alpha-GFN to generalize GFN training. It provides both theoretical and empirical results to demonstrate its effectiveness.

Weakness: The paper presentation needs improvement and the mathematical content needs to be better presented. The experimental results need to be better presented and explained. The technical novelty also needs to be better highlighted.

**Reviewer Concerns:**

Weakness: The paper presentation needs improvement and the mathematical content needs to be better presented. The experimental results need to be better presented and explained. The technical novelty also needs to be better highlighted.

**Reviewer Scores:**

Based on the comments and discussions, the reviewers will likely keep their scores.

---

### Decision · Program_Chairs · 2026-01-26

Reject